# Convergence of topological domain boundaries, insulators, and polytene interbands revealed by high-resolution mapping of chromatin contacts in the early *Drosophila melanogaster* embryo

Michael R Stadler[1], Jenna E Haines[1], Michael B Eisen[1,2,3]*

[1]Department of Molecular and Cell Biology, University of California, Berkeley, CA, United States; [2]Department of Integrative Biology, University of California, Berkeley, CA, United States; [3]Howard Hughes Medical Institute, Berkeley, CA, United States

**Abstract** High-throughput assays of three-dimensional interactions of chromosomes have shed considerable light on the structure of animal chromatin. Despite this progress, the precise physical nature of observed structures and the forces that govern their establishment remain poorly understood. Here we present high resolution Hi-C data from early *Drosophila* embryos. We demonstrate that boundaries between topological domains of various sizes map to DNA elements that resemble classical insulator elements: short genomic regions sensitive to DNase digestion that are strongly bound by known insulator proteins and are frequently located between divergent promoters. Further, we show a striking correspondence between these elements and the locations of mapped polytene interband regions. We believe it is likely this relationship between insulators, topological boundaries, and polytene interbands extends across the genome, and we therefore propose a model in which decompaction of boundary-insulator-interband regions drives the organization of interphase chromosomes by creating stable physical separation between adjacent domains.

DOI: https://doi.org/10.7554/eLife.29550.001

*For correspondence:
mbeisen@berkeley.edu

**Competing interests:** The authors declare that no competing interests exist.

## Introduction

Beginning in the late 19th century, cytological investigations of the polytene chromosomes of insect salivary glands implicated the physical structure of interphase chromosomes in their cellular functions (*Balbiani, 1881*; *Balbiani, 1890*; *Heitz and Bauer, 1933*; *King and Beams, 1934*; *Painter, 1935*). Over the next century plus, studies in the model insect species *Drosophila melanogaster* were instrumental in defining structural features of animal chromatin. Optical and electron microscopic analysis of fly chromosomes produced groundbreaking insights into the physical nature of genes, transcription and DNA replication (*Benyajati and Worcel, 1976*; *Laird and Chooi, 1976a*; *Laird et al., 1976b*; *McKnight and Miller, 1976*; *McKnight and Miller, 1977*; *Vlassova et al., 1985*; *Belyaeva and Zhimulev, 1994*).

Detailed examination of polytene chromosomes in *Drosophila melanogaster* revealed a stereotyped organization, with compacted, DNA-rich 'bands' alternating with extended, DNA-poor 'interband' regions (*Bridges, 1934*; *Rabinowitz, 1941*; *Lefevre, 1976*; *Benyajati and Worcel, 1976*; *Laird and Chooi, 1976a*), and it appears likely that this structure reflects general features of chromatin organization shared by non-polytene chromosomes. While these classical studies offered

**eLife digest** The DNA inside a cell is packaged into threaded structures called chromosomes. Early studies of chromosomes using insect larvae revealed a pattern of dark- and light-colored bands on the chromosomes that was unique to every region. For decades, it remained unclear if the bands had a specific role.

More advanced techniques have shown that chromosomes are organized into a series of compact domains that contain separate regions of genes. Each region can be turned on or off at different times, depending on the needs of different cells. This allows cells to specialize into different cell types and tissues. Until now, it was unclear how these different regions are formed and controlled, and how they relate to the chromosome bands.

Here, Stadler, Haines and Eisen used a specific technique to map the structure of chromosomes in early fly embryos, by using a chemical trap to capture closely located DNA pieces. The results showed that the chromosome domains corresponded to the banding patterns seen in the early studies. This suggest that light bands represent extended DNA that act as spacers between the dark gene regions.

This study adds to the view that the way the DNA is organized influences gene activity. Creating a high-resolution model of the chromosomes will help us to better understand how their structure can influence the activity of genes. In future, scientists may be able to identify diseases that are caused by errors in the chromosome organization.

DOI: https://doi.org/10.7554/eLife.29550.002

extensive structural and molecular characterization of chromosomes in vivo, the question of what was responsible for organizing chromosome structure remained unanswered.

A critical clue came with the discovery of insulators, DNA elements initially identified based on their ability to block the activity of transcriptional enhancers when located between an enhancer and its targeted promoters (*Kellum and Schedl, 1991*; *Holdridge and Dorsett, 1991*; *Geyer and Corces, 1992*; *Kellum and Schedl, 1992*). Subsequent work showed that these elements could also block the spread of silenced chromatin states (*Roseman et al., 1993*; *Sigrist and Pirrotta, 1997*; *Mallin et al., 1998*; *Recillas-Targa et al., 2002*; *Kahn et al., 2006*) and influence the structure of chromatin. Through a combination of genetic screens and biochemical purification, a number of protein factors have been identified that bind to *Drosophila* insulators and modulate their function, including Su(Hw), BEAF-32, mod(mdg4), CP190, dCTCF, GAF, Zw5, and others (*Lindsley and Grell, 1968*; *Lewis, 1981*; *Parkhurst and Corces, 1985*; *Parkhurst and Corces, 1986*; *Spana et al., 1988*; *Parkhurst et al., 1988*; *Zhao et al., 1995*; *Gerasimova et al., 1995*; *Bell et al., 1999*; *Gaszner et al., 1999*; *Scott et al., 1999*; *Büchner et al., 2000*; *Pai et al., 2004*; *Melnikova et al., 2004*; *Moon et al., 2005*). Except for CTCF, which is found throughout bilateria, all of these proteins appear to be specific to arthropods (*Heger et al., 2013*).

Staining of polytene chromosomes with antibodies against such insulator proteins showed that many of them localize to polytene interbands (*Belyaeva and Zhimulev, 1994*; *Zhao et al., 1995*; *Byrd and Corces, 2003*; *Eggert et al., 2004*; *Pai et al., 2004*; *Gortchakov et al., 2005*; *Gilbert et al., 2006*; *Berkaeva et al., 2009*; *Vatolina et al., 2011b*), with some enriched at interband borders. Further, some, though not all, insulator protein mutants disrupt polytene chromosome structure (*Roy et al., 2007*). Together, these data implicate insulator proteins, and the elements they bind, in the organization of the three-dimensional structure of fly chromosomes.

Several high-throughput methods to probe three-dimensional structure of chromatin have been developed in the last decade (*Lieberman-Aiden et al., 2009*; *Fullwood et al., 2009*; *Rao et al., 2014*; *Beagrie et al., 2017*). Principle among these are derivatives of the chromosome conformation capture (3C) assay (*Dekker et al., 2002*), including the genome-wide 'Hi-C' (*Lieberman-Aiden et al., 2009*). Several groups have performed Hi-C on *Drosophila* tissues or cells and have shown that fly chromosomes, like those of other species, are organized into topologically associated domains (TADs), regions within which loci show enriched 3C linkages with each other but depleted linkages with loci outside the domain. Disruption of TAD structures by gene editing in mammalian

cells has been shown to disrupt enhancer-promoter interactions and significantly alter transcriptional activity (*Guo et al., 2015*; *Lupiáñez et al., 2015*).

Although TADs appear to be a common feature of animal genomes, the extent to which TAD structures are a general property of a genome or if they are regulated as a means to control genome function remains unclear, and the question of how TAD structures are established remains largely open. Previous studies have implicated a number of features in the formation of *Drosophila* TAD boundaries, including transcriptional activity and gene density, and have reached differing conclusions about the role played by insulator protein binding (*Sexton et al., 2012*; *Hou et al., 2012*; *Van Bortle et al., 2014*; *Ulianov et al., 2016*; *Li et al., 2015*). Tantalizingly, Eagen et al., using 15 kb resolution Hi-C data from *D. melanogaster* have shown that there is a correspondence between the distribution of large TADs and polytene bands (*Eagen et al., 2015*).

We have been studying the formation of chromatin structure in the early *D. melanogaster* embryo because of its potential impact on the establishment of patterned transcription during the initial stages of development. We have previously has shown that regions of 'open' chromatin are substantially remodeled at enhancers and promoters during early development (*Harrison et al., 2011*; *Li et al., 2014*) and were interested in the role three-dimensional chromatin structure plays in spatial patterning.

We therefore generated high-resolution Hi-C datasets derived from nuclear cycle 14 *Drosophila melanogaster* embryos (*Foe and Alberts, 1983*), and from the anterior and posterior halves of hand-dissected embryos at the same developmental stage. We show that high-resolution chromatin maps of anterior and posterior halves are nearly identical, suggesting that chromatin structure neither drives nor directly reflects spatially patterned transcriptional activity. However, we show that stable long-range contacts evident in our chromatin maps generally involve known patterning genes, implicating chromatin conformation in transcriptional regulation.

To investigate the origins of three-dimensional chromatin structure, we carefully map the locations of the boundaries between topological domains using a combination of manual and computational annotation. We demonstrate that these boundaries resemble classical insulators: short (500–2000 bp) genomic regions that are strongly bound by (usually multiple) insulator proteins and are sensitive to DNase digestion. Additionally, we find that boundaries share the molecular features of polytene interband regions. Finally, we show that for a region in which the fine polytene banding pattern has been mapped to genomic positions, boundaries show precise colocalization with interband regions that separate compacted bands corresponding to TADs. We propose that this relationship between insulators, TADs and polytene interbands extends across the genome, and suggest a model in which the decompaction of these regions drives the organization of interphase fly chromosomes by creating stable physical separation between adjacent domains.

## Results

### Data quality and general features

We prepared and sequenced in situ Hi-C libraries from two biological replicates of hand-sorted cellular blastoderm (mitotic cycle 14; mid-stage 5) embryos using a modestly adapted version of the protocol described in *Rao et al., 2014*. To examine possible links between chromatin maps and transcription, we sectioned hand-sorted mitotic cycle 14 embryos along the anteroposterior midline, and generated Hi-C data from the anterior and posterior halves separately, also in duplicate. In total, we produced ~452 million informative read pairs (see *Supplementary file 1*).

We assessed the quality of these data using metrics similar to those described by (*Lieberman-Aiden et al., 2009*; *Rao et al., 2014*). Specifically, the strand orientations of our reads were approximately equal in each sample (as expected from correct Hi-C libraries but not background genomic sequence; see *Supplementary file 1*), the signal decay with genomic distance was similar across samples, and, critically, visual inspection of heat maps prepared at a variety of resolutions showed these samples to be very similar both to each other and to previously published data prepared using similar methods (*Sexton et al., 2012*). We conclude that these Hi-C are of high quality and reproducibility.

We next sought to ascertain the general features of the data at low resolution. We examined heatmaps for all *D. melanogaster* chromosomes together using 100 kb bins, as shown in *Figure 1*.

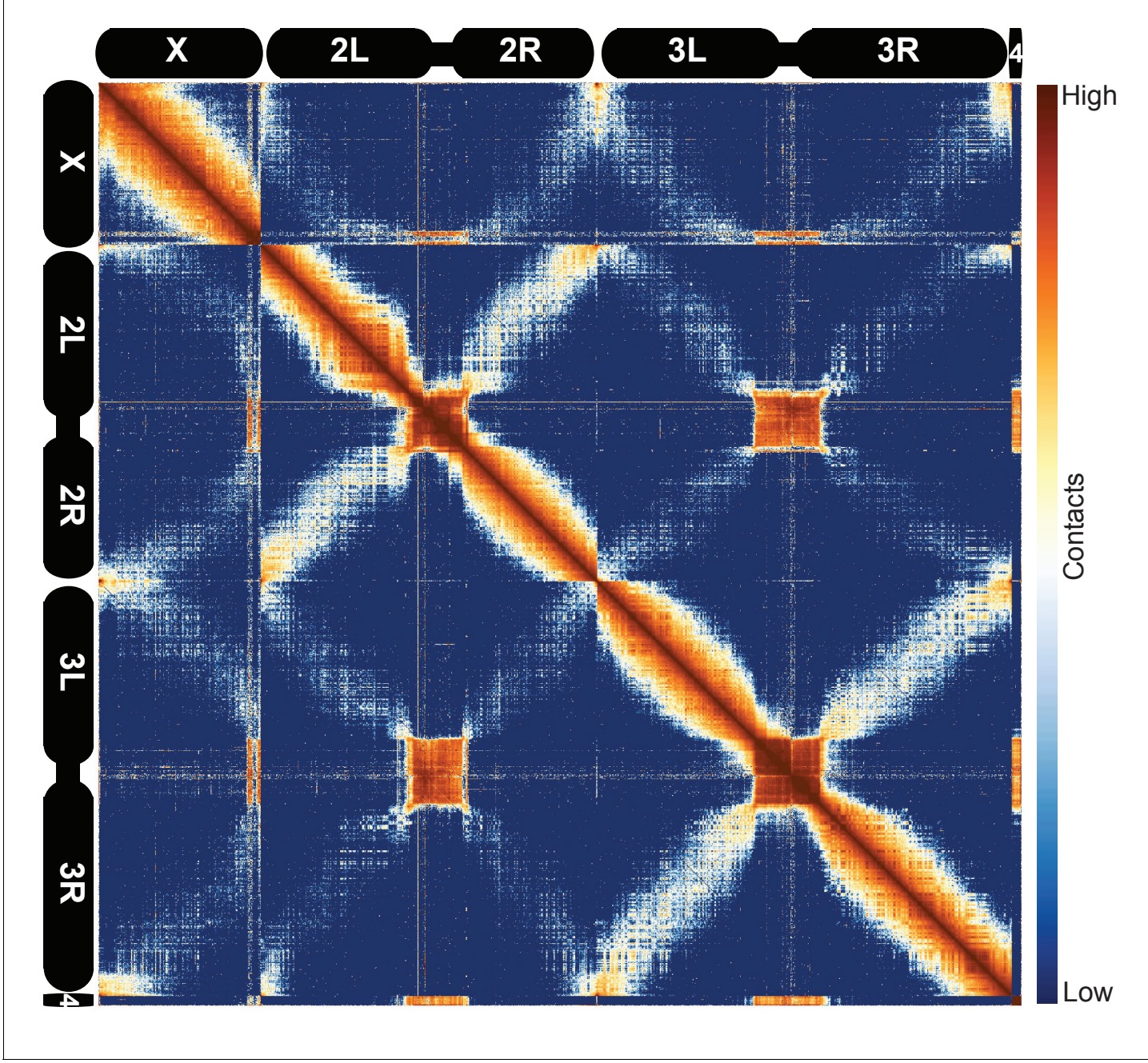

**Figure 1.** Hi-C map of the stage 5 *Drosophila melanogaster* genome at 100 kb resolution. Data from all nc14 datasets was aggregated and normalized by the 'vanilla coverage' method. To enhance contrast, the logarithm values of the normalized counts were histogram equalized, and maximum and minimum values were adjusted for optimal display.

DOI: https://doi.org/10.7554/eLife.29550.003

Several features of the data are immediately apparent. The prominent 'X' patterns for chromosomes 2 and 3, which indicate an enrichment of linkages between chromosome arms, reflects the known organization of fly chromosomes during early development known as the Rabl configuration (*Csink and Henikoff, 1998*; *Wilkie et al., 1999*; *Duan et al., 2010*): telomeres are located on one side of the nucleus, centromeres are located on the opposite side, and chromosome arms are arranged roughly linearly between them. Centromeres and the predominantly heterochromatic chromosome 4 cluster together, as, to a lesser extent, do telomeres, reflecting established cytological

features that have been detected by prior Hi-C analysis (*Sexton et al., 2012*) and fluorescence in situ hybridization (FISH) (*Lowenstein et al., 2004*). These features were evident in all replicates, further confirming both that these datasets are reproducible and that they capture known features of chromatin topology and nuclear arrangement.

## TAD boundaries are short elements bound by insulator proteins

Because we used a 4-cutter restriction enzyme and deep sequencing, and because the fly genome is comparatively small, we were able to resolve features at high resolution. We visually inspected genome-wide maps of a number of genomic regions constructed using bins of 500 bp, and were able to see a conspicuous pattern of TADs across a wide range of sizes, some smaller than 5 kb (*Figure 2*, *Figure 2—figure supplements 1–5*). When we compared maps for several of these regions with available functional genomic data from embryos, we observed that the boundaries between these domains showed a remarkably consistent pattern: they were formed by short regions of DNA (500–2000 bp) that are nearly always associated with high chromatin accessibility, measured by DNase-seq (*Li et al., 2011*), strong occupancy by known insulator proteins as measured by chromatin immunoprecipitation (ChIP) (*Nègre et al., 2010*) (*Figure 2*, *Figure 2—figure supplements 1–5*) properties characteristic of classical *Drosophila* insulator elements.

To confirm this visually striking association, we systematically called TAD boundaries by visual inspection of panels of raw Hi-C data covering the entire genome. Critically, these boundary calls were made from Hi-C data alone, and the human caller lacked any information about the regions being examined, including which region (or chromosome) was represented by a given panel. In total, we manually called 3122 boundaries in the genome for nc14 embryos. Taking into account the ambiguity associated with intrinsically noisy data, the difficulty of resolving small domains, and the invisibility of sections of the genome due to repeat content or a lack of MboI cut sites, we consider 4000–5500 to be a reasonable estimate for the number of boundaries in the genome.

To complement these manual calls, we developed a computational approach for calling boundaries that is similar to methods used by other groups (*Lieberman-Aiden et al., 2009*; *Sexton et al., 2012*; *Rao et al., 2014*; *Crane et al., 2015*). In brief, we assigned a directionality score to each genomic bin based on the number of Hi-C reads linking the bin to upstream versus downstream regions, and then used a set of heuristics to identify points of transition between regions of upstream and downstream bias. We adjusted the parameters of the directionality score and the boundary calling to account for features of the fly genome, specifically the relatively small size of many topological domains.

Attempts to exhaustively and definitively identify features within genomic data are necessarily variable due to differences in the choice of algorithm, parameters, cutoffs, and unavoidable tradeoffs between sensitivity and accuracy. We therefore sought a representative set of TAD boundaries with which to analyze features of these elements. Of our top 1000 computationally-identified domain boundaries, we found that 952 were matched by a manually-called boundary within 1 kb. This high level of agreement suggested that the computational approach robustly identified the domain features that are apparent by eye. By taking the union of our computational calls, applied with a stringent cutoff, and our manual calls, we developed a very conservative set of exceptionally high confidence boundaries. We emphasize that this set represents only a subset of the boundaries identified by manual and computational approaches, the complete lists of which are provided in *Supplementary file 1*.

Comparing these 952 boundaries to other genomic datasets confirms our initial observations and reveals a highly stereotyped pattern of associated genomic features. Most strikingly, boundaries are enriched for the binding of the known insulator proteins CP190, BEAF-32, mod(mdg4), dCTCF, and to a lesser extent GAF and Su(Hw) (*Figure 3*). CP190 and BEAF-32 show the strongest enrichment, and indeed, virtually all (95.1%) of the examined boundaries appear to be associated with CP190 binding (*Figure 3—figure supplement 1*). Domains of H3K27 trimethylation, a marker of polycomb silencing, showed a strong tendency to terminate at boundaries, and the enhancer mark H3K4me1 showed an interesting pattern of depletion at boundaries but enrichment immediately adjacent to boundary locations (*Figure 3*). Boundaries also exhibit peaks of DNase accessibility and nucleosome depletion (*Figure 3*), as well as marks associated with promoters, including the general transcription factors TFIIB and the histone tail modification H3K4me3. Despite the presence of promoter marks,

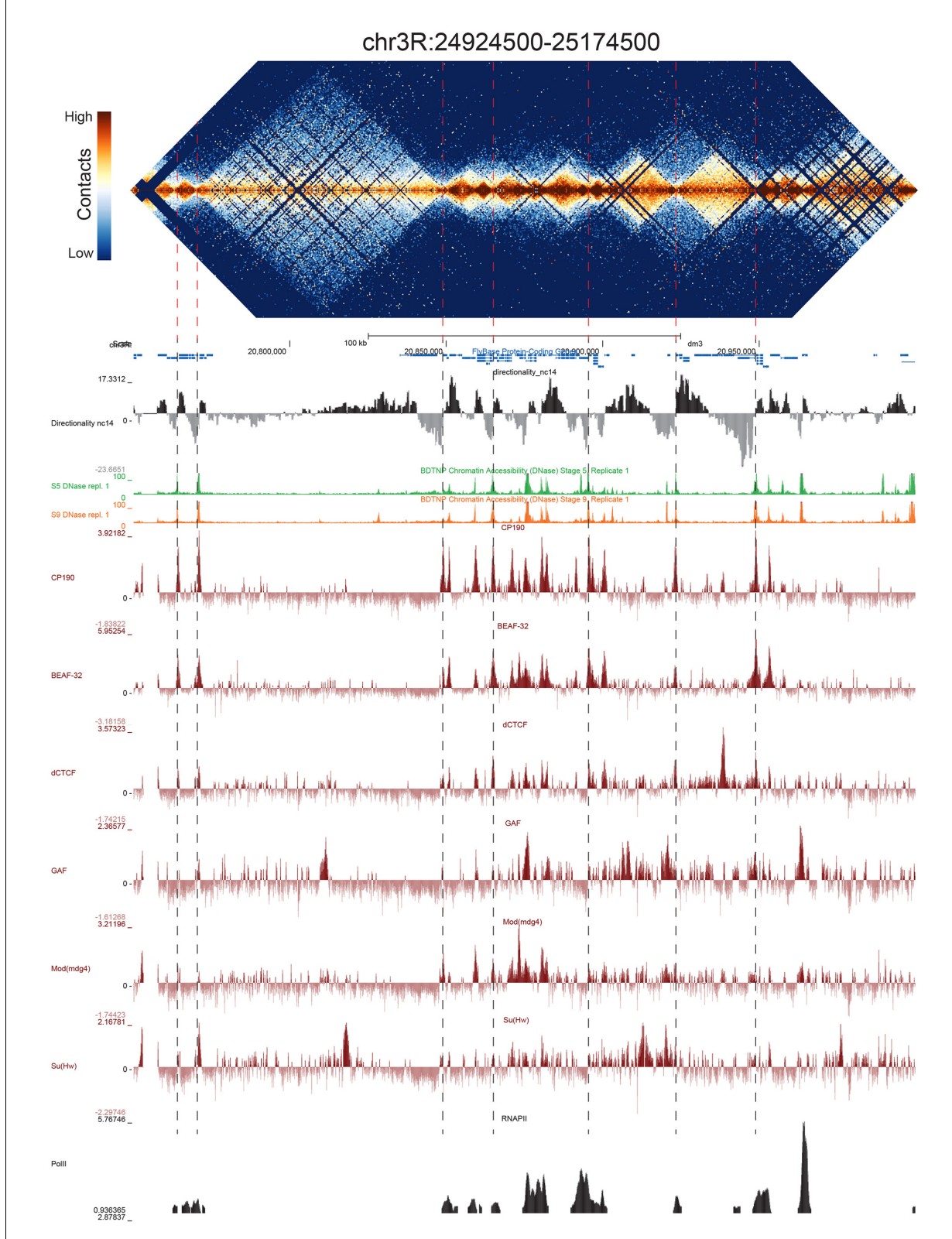

**Figure 2.** Example region of Hi-C data at 500 bp resolution. Heat map of aggregate Hi-C data for all nc14 datasets binned at 500 bp is shown for the region located at 3R:24924500–25174500 (dm3: 3R:20750000–20999999). Raw counts were normalized by the vanilla coverage method, the logarithm was taken, and minimum and maximum values were adjusted for visual contrast. A UCSC browser (*Kent et al., 2002*) window for the corresponding coordinates is shown with tracks for Hi-C directionality (calculated from the Hi-C data shown in the heatmap), DNase accessibility (X.-Y. *Li et al., 2011*),

*Figure 2 continued on next page*

*Figure 2 continued*

RNA polII and TFIIB (*Li et al., 2008*), and the insulator proteins CP190, BEAF-32, dCTCF, GAF, mod(mdg4), and Su(Hw) from (*Nègre et al., 2010*). Dashed red lines are visual guides and are manually drawn at locations of apparent boundaries; they do not reflect algorithmically or unbiased hand-curated boundary calls.

DOI: https://doi.org/10.7554/eLife.29550.004

The following figure supplements are available for figure 2:

**Figure supplement 1.** High resolution Hi-C maps of additional example genomic regions from stage 5 *Drosophila melanogaster* embryos.

DOI: https://doi.org/10.7554/eLife.29550.005

**Figure supplement 2.** High resolution Hi-C maps of additional example genomic regions from stage 5 *Drosophila melanogaster* embryos.

DOI: https://doi.org/10.7554/eLife.29550.006

**Figure supplement 3.** High resolution Hi-C maps of additional example genomic regions from stage 5 *Drosophila melanogaster* embryos.

DOI: https://doi.org/10.7554/eLife.29550.007

**Figure supplement 4.** High resolution Hi-C maps of additional example genomic regions from stage 5 *Drosophila melanogaster* embryos.

DOI: https://doi.org/10.7554/eLife.29550.008

**Figure supplement 5.** High resolution Hi-C maps of additional example genomic regions from stage 5 *Drosophila melanogaster* embryos.

DOI: https://doi.org/10.7554/eLife.29550.009

we find that RNA polII is present at only a subset (45.1%) of stage 5 boundaries (*Figure 3*, *Figure 3—figure supplement 1*).

It is striking that we observe that not only are sites of combinatorial insulator protein binding enriched at TAD boundaries, but they are highly predictive. Of our representative set of boundaries, 95.1% are are enriched >2 fold for CP190 binding within a 1.5 kb window. Conversely, of the strongest 1000 CP190 peaks, 75.2% are within 2 kb of a manual or computationally-called boundary (compared to 37.4% of the top 1000 RNAPII peaks). It is important to note that we do identify a small subset of boundaries that are not apparently associated with sites of insulator binding (~1–2% show no enrichment for CP190, BEAF-32, or dCTCF, depending on thresholds used), suggesting that there are multiple phenomena that can create topological boundaries in flies (e.g., see Figure 6). However, the overwhelming majority of topological boundaries identified in this study coincide with elements that match the properties of CP190-associated insulators.

An important confounding factor in sorting out the nature of topological boundaries is the strong tendency, observed by multiple authors, of insulator proteins to bind near promoters specifically between divergent promoters (*Nègre et al., 2010*; *Ramirez et al., 2017*; *Schwartz et al., 2012*). Indeed, we find that boundary elements, as identified from Hi-C, are often found proximal to promoters and show a general enrichment of promoter-associated marks (*Figure 3*), raising the possibility that transcriptional activity at promoters may drive topological boundary formation. However, a number of features of the data argue against this possibility. First and most critically, many of the topological boundaries (54.9%) we identify are not associated with RNAPII binding in nc14 embryos. Similarly, there are many active promoters that do not appear to form topological boundaries (e.g., Figure 8 and supplements). *Hug et al., 2017* pharmacologically inhibited transcription in early embryos and observe that TADs remain intact. Finally, topological boundaries are invariant between anterior and posterior sections of embryos despite substantial differences in the transcriptional profiles of these regions (see below). We further examined the distributions of the same genomic features around the top 1000 peaks of H3K4me3, a marker of active promoters, in data from stage 5 embryos (*Figure 3—figure supplement 2*) (*Li et al., 2014*). While these sites show enrichments for insulator proteins, these enrichments are substantially weaker than those observed at topological boundaries, while RNA polII enrichment is much stronger at promoters than boundaries. The tendency for polycomb domains to terminate at promoters is also much less pronounced at promoters than boundaries. Together, these data argue that boundaries constitute a distinct class of genetic elements that are not formed by promoter transcription, but are instead frequently located near promoters, possibly as a result of selective pressure to insulate these proximal promoters from distal regulatory elements. While we cannot rule out any role for promoter-bound transcription machinery in the formation of topological boundaries (notably, TFIIB is enriched at 69.1% of boundaries), we think it is unlikely that transcriptional activity plays a major role in establishing the topological domains of interphase fly chromosomes.

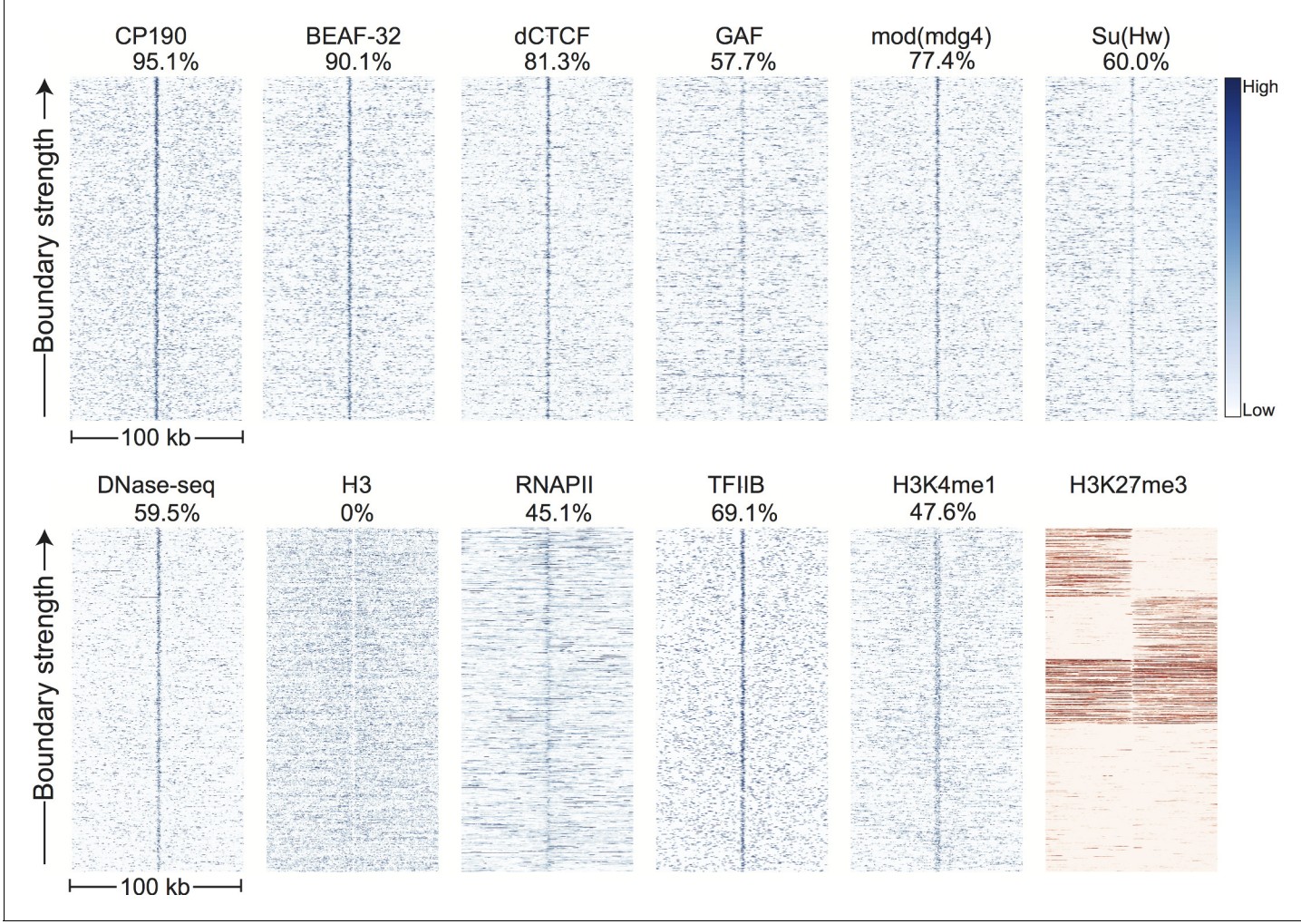

**Figure 3.** Topological domain boundaries show distinct patterns of associated proteins and genomic features. Heatmaps showing the distribution of signals from embryonic ChIP and DNase-seq datasets around 952 topological boundaries identified jointly by computational and manual curation. All plots show 500 bp genomic bins in 100 kb windows around boundaries. All plots in blue are sorted by boundary strength, calculated from the difference in upstream and downstream Hi-C directionality scores. The plot for H3K27me3 (in red) is specially sorted to highlight the tendency for enriched domains to terminate at boundaries. Rows for this plot were sorted by calculating the total H3K27me3 signal in the 50 kb windows upstream and downstream of the boundary and then sorting, top to bottom: upstream signal above median and downstream signal below the median, upstream below and downstream above, upstream and downstream both above, upstream and downstream both below the median. For comparison, identically prepared and sorted plots around H3K4me3 peaks are shown in *Figure 3—figure supplement 2*. Percentages are calculated as the percentage of boundaries with a >2 fold enrichment for the given signal within a 3 kb window centered on the boundary (±1.5 kb). Data for insulator proteins, DNase accessibility, RNA polII and TFIIB are from the same sources indicated in *Figure 2*. ChIP for H3, H3K4me1 are taken from (*Li et al., 2014*), and H3K27me3 are from modEncode (*Contrino et al., 2012*).

DOI: https://doi.org/10.7554/eLife.29550.010

The following figure supplements are available for figure 3:

**Figure supplement 1.** Genomic signals around topological boundaries, self-sorted.
DOI: https://doi.org/10.7554/eLife.29550.011
**Figure supplement 2.** Genomic signals around H3K4me3 peaks.
DOI: https://doi.org/10.7554/eLife.29550.012
**Figure supplement 3.** Directionality around peaks of genomic features.
DOI: https://doi.org/10.7554/eLife.29550.013
**Figure supplement 4.** Genomic signals around H3K4me1 peaks.
DOI: https://doi.org/10.7554/eLife.29550.014

Finally, we examined the sequence composition of boundary elements by comparing the frequency of DNA words of up to seven base pairs in the set of high confidence boundaries to flanking sequence. The most enriched sequences correspond to the known binding site of BEAF-32 and to a CACA-rich motif previously identified as enriched in regions bound by CP190 (*Nègre et al., 2010*; *Yang and Corces, 2012*), both of which show strong association with the set of boundary sequences as a whole (*Figure 4*).

## Boundary elements correspond to polytene interbands

The examination of these boundary elements led us to consider the physical basis of topological domain separation. Chromosome conformation capture is a complex assay (*Gavrilov et al., 2013*; *Gavrilov et al., 2015*), and inferring discrete physical states of the chromatin fiber from Hi-C signals generally requires orthogonal experimental data. To address this problem, we sought to leverage information from polytene chromosomes to draw associations between features of Hi-C data and physical features of chromosomes.

The Zhimulev laboratory has extensively studied the nature and composition of polytene bands and interbands for decades. Using a combination of approaches, they have identified interbands as a set of ~5700 short decompacted regions that tend to be located near divergent promoters and are characterized by DNase hypersensitivity and the binding of characteristic proteins, including insulator proteins (*Zhimulev et al., 2014*). It was immediately apparent to us that these elements bore significant similarity to the topological boundary elements we identified. We thus sought to compare our Hi-C data to known polytene chromosome structures.

There is surprisingly little data mapping features of polytene structure to specific genomic coordinates at high resolution. *Vatolina et al., 2011a* used exquisitely careful electron microscopy to identify the fine banding pattern of the 65 kb region between polytene bands 10A1-2 and 10B1-2, revealing that this region, which appears as a single interband under a light microscope, actually contains six discrete, faint bands and seven interbands. The region is flanked by two large bands, whose genomic locations have been previously mapped and refined by FISH (*Vatolina et al., 2011a*). Vatolina et al. then used available molecular genomic data to propose a fine mapping of these bands and interbands to genomic coordinates.

*Figure 5* shows the correspondence between Vatolina et al.'s proposed polytene map from this region and our high-resolution Hi-C data, along with measures of early embryonic DNase hypersensitivity from (*Li et al., 2011*) and the binding of six insulator proteins (*Nègre et al., 2010*). There is a striking correspondence between the assignments of Vatolina et al. and our Hi-C data: faint polytene bands correspond to TADs, and interbands correspond to the boundary elements that separate the TADs.

This correspondence is not perfect. Specifically, the evidence in our Hi-C data for the separation between the major band 10A1-2 and the minor band 10A3 is weak, though that may be partly explained by the absence of MboI cut sites obscuring much of this region. This minor band is barely detectable in polytene spreads, and the combination of this and weak support in Hi-C data may suggest that this band is not real or is perhaps only present in a minority of nuclei. Similarly, the Vatolina et al. report that they only rarely observe the interband between bands 10A6 and 10A7, and we indeed observe substantial contact between these two putative bands in Hi-C maps (the light orange region near the peak of the 'pyramid' formed by 10A6 and 10A7 in *Figure 5*), though each shows stronger intra- than inter-domain interactions. One possible explanation for this observation is that the interband separating these two domains is not constitutive but rather is formed in only a fraction of nuclei. The pattern exhibited by these two domains–adjacent domains that show a clear separation but also a substantial interaction signal–is one we observe frequently in our early embryonic Hi-C data, suggesting that variable boundaries may be common features of the fly genome.

Overall, the alignment between polytene band mapping and Hi-C data in this region supports a strong correspondence between these two types of data. For five interbands which were easily visible in polytene spreads (10A3/4-5, 10A4-5/10A6, 10A7/10A8-9, 10A8-9/10A10, 10A10-11/10B1), we observe strong domain boundaries in Hi-C data. For two interbands supported by weaker evidence in polytene analysis, we observe in Hi-C maps a weak or non-existent boundary (10A1-2/10A3) and a boundary with significant interaction across it, possibly representing heterogeneity between nuclei matching heterogeneity in polytenes (10A6/10A7).

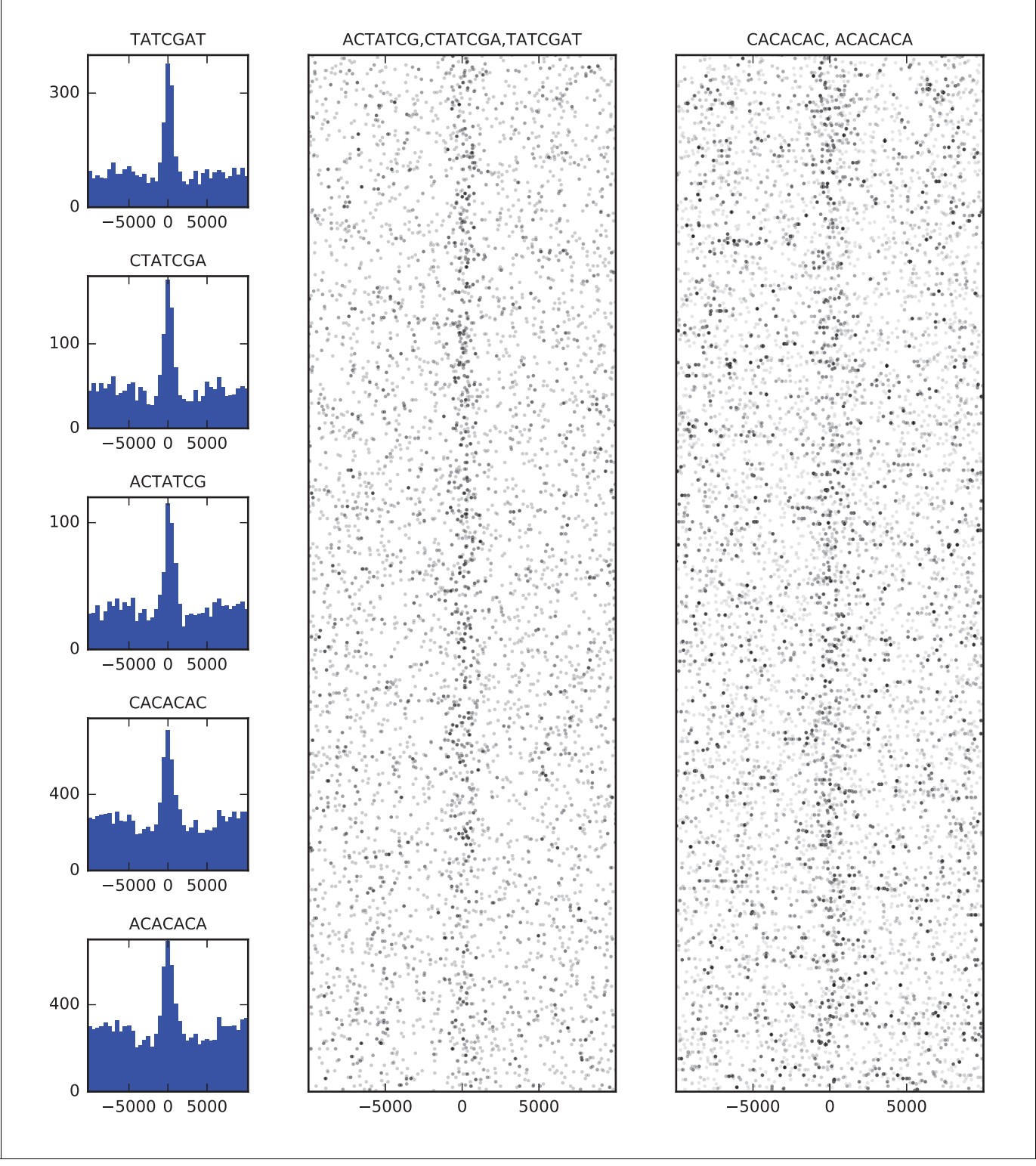

**Figure 4.** Sequence features of TAD boundary elements. (**A**) Histograms showing the frequency of enriched 7-mers in 5 kb windows around 952 high-confidence TAD boundaries. (**B**) Scatter plots of occurrences of words matching known BEAF-32 binding motifs (left) and CACA motif (right) in 10 kb windows around high-confidence TAD boundaries. Points are plotted with low opacity, such that darker points correspond to positions where multiple words occurs close together in sequence.

DOI: https://doi.org/10.7554/eLife.29550.015

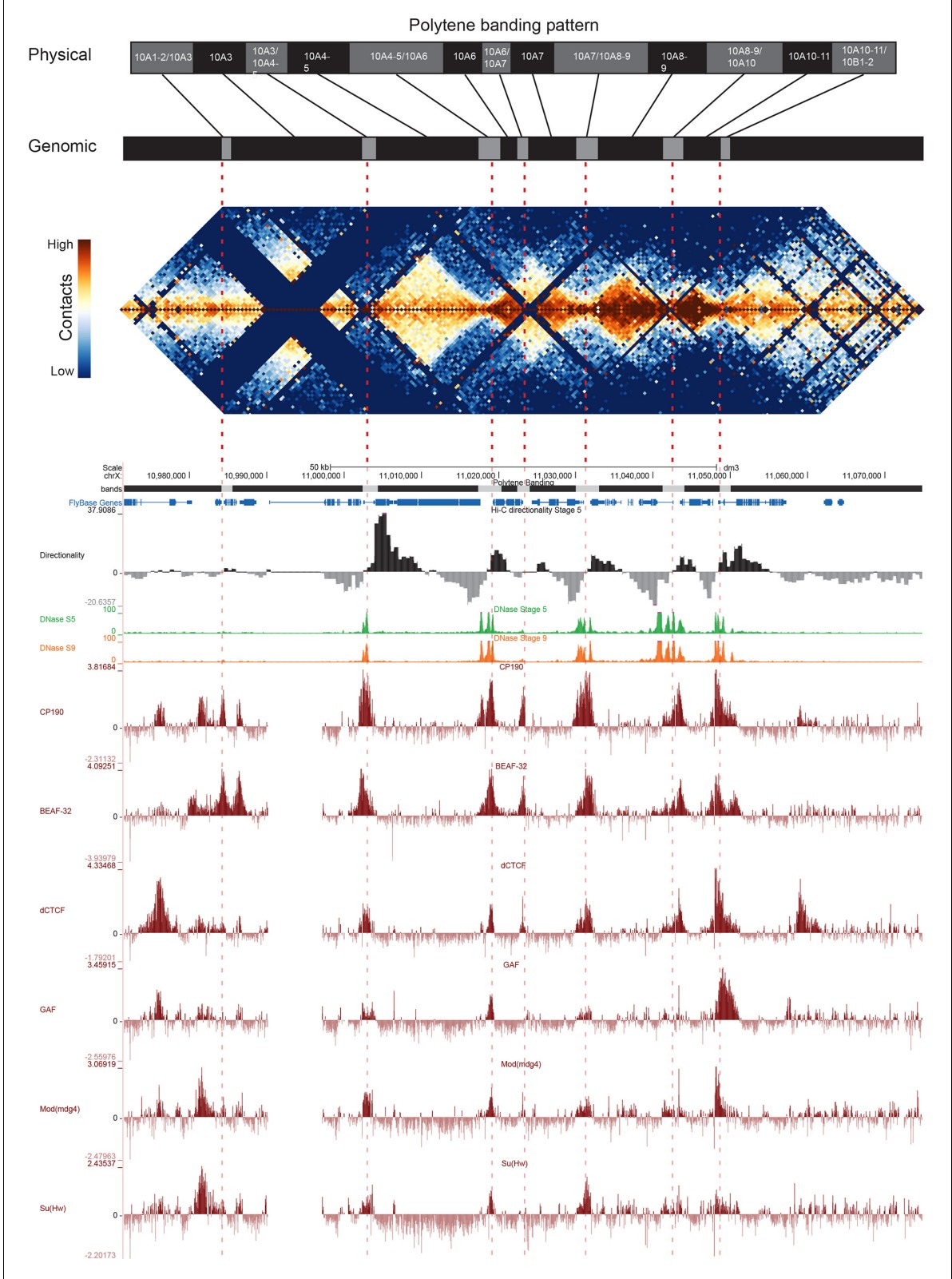

**Figure 5.** Topological boundary elements correspond to polytene interbands. Heat map of aggregate Hi-C data for all nc14 datasets binned at 500 bp and UCSC browser data shown for the region X:11077500–11181000 (dm3: X:10971500–11075000) for which Vatolina et al. provided fine-mapping of polytene banding structure. Hi-C and browser data were prepared and sourced as indicated in *Figure 2*. Dashed red lines are visual guides drawn from the interband assignments of Vatolina et al. Top: accurately-scaled representations of the size of the mapped bands and interbands in base pairs

*Figure 5 continued on next page*

*Figure 5 continued*

('Genomic') and the corresponding physical distances in polytene chromosomes derived from electron microscopic analysis of polytene chromosomes by Vatolina et al. Increased relative physical size of interband regions demonstrates their lower compaction ratios.

DOI: https://doi.org/10.7554/eLife.29550.016

The following figure supplements are available for figure 5:

**Figure supplement 1.** TAD structure corresponds to mapped polytene structure at the *Notch* locus.

DOI: https://doi.org/10.7554/eLife.29550.017

**Figure supplement 2.** Chriz protein binding in Kc167 cells is highly enriched at sites of embryonic nc14 topological boundaries.

DOI: https://doi.org/10.7554/eLife.29550.018

The 5' region of the *Notch* gene has also been carefully mapped. Rykowski et al. used high-resolution in situ hybridization to determine that the coding sequences of *Notch* lies within polytene band 3C7, while the sequences upstream of the transcription start site (TSS) lie in the 3C6-7 interband. Examining the *Notch* locus in our Hi-C data, we see that the gene body is located within an ~20 kb TAD, and the TSS directly abuts a TAD boundary that is strongly bound by CP190 and dCTCF (*Figure 5—figure supplement 1*), an arrangement consistent with the correspondence of boundaries and interbands.

The chromodomain-containing protein Chriz has been suggested as the strongest diagnostic feature of polytene interbands (*Zhimulev et al., 2014*). Using publicly available ChIP datasets from Kc167 cells (derived from late embryonic tissue), we observed a strong enrichment of Chriz binding at our representative boundaries (87.9% >2 fold enriched within 1.5 kb, *Figure 5—figure supplement 2A*). Further, Hi-C directionality around Chriz peaks shows the characteristic pattern of boundary formation, and Chriz profiles at representative loci show substantial correspondence between boundary regions and Chriz binding (*Figure 5—figure supplement 2B and C*), offering further support for the association between boundaries and interbands.

Eagen et al. previously identified a broad correspondence between polytene interbands and inter-TAD regions from Hi-C data at 15 kb resolution (*Eagen et al., 2015*). Our Hi-C data allows the detection of fine structure within these inter-TAD regions, down to individual boundary elements. Owing to the dearth of finely mapped polytene regions, the association between topological boundaries and interband regions is necessarily based on a limited number of example loci. However, the combination of data from these loci with the close agreement of the molecular composition of these regions, specifically the strong localization of the interband marker Chriz to topological boundaries, leads us to propose that the precise relationship between topological boundaries, insulator elements, and decompacted interband regions we observe is a general one, and that it extends across the genome.

The association between boundary elements and interbands suggests a simple model for insulator function. A key feature that distinguishes polytene interbands from bands is their low compaction ratio: they span a larger physical distance per base pair. The association between insulator binding and genomic regions with low compaction ratios suggests insulators may function by simply increasing the physical distance between adjacent domains via the unpacking and extension of intervening chromatin. *Figure 5* (top) shows a representation of the conversion of genomic distance to physical distance for the 10A1-B1 region, as measured by Vatolina et al. Any model for insulator function must explain several features of insulator function, including the ability to organize chromatin into physical domains, block interactions between enhancers and promoters exclusively when inserted between them, protect transgenes from position effect variegation and block the spread of chromatin silencing states. This chromatin extension model for fly insulator function can potentially explain these defining characteristics via simple physical separation.

## Hi-C data can elaborate fine polytene structures

We reasoned that if our Hi-C data is capable of resolving fine banding patterns such as that at the 10A1-B1 locus, we should be able to resolve the borders of major bands with precision. We focused on a region of chromosome 2L that had previously been shown by Eagen et al. to appear as a single ~500 kb TAD using Hi-C at 15 kb resolution, but contains a faint interband in Bridge's map. Our Hi-C data reveal an intricate structure at this locus (*Figure 6A*). There are two large TADs on

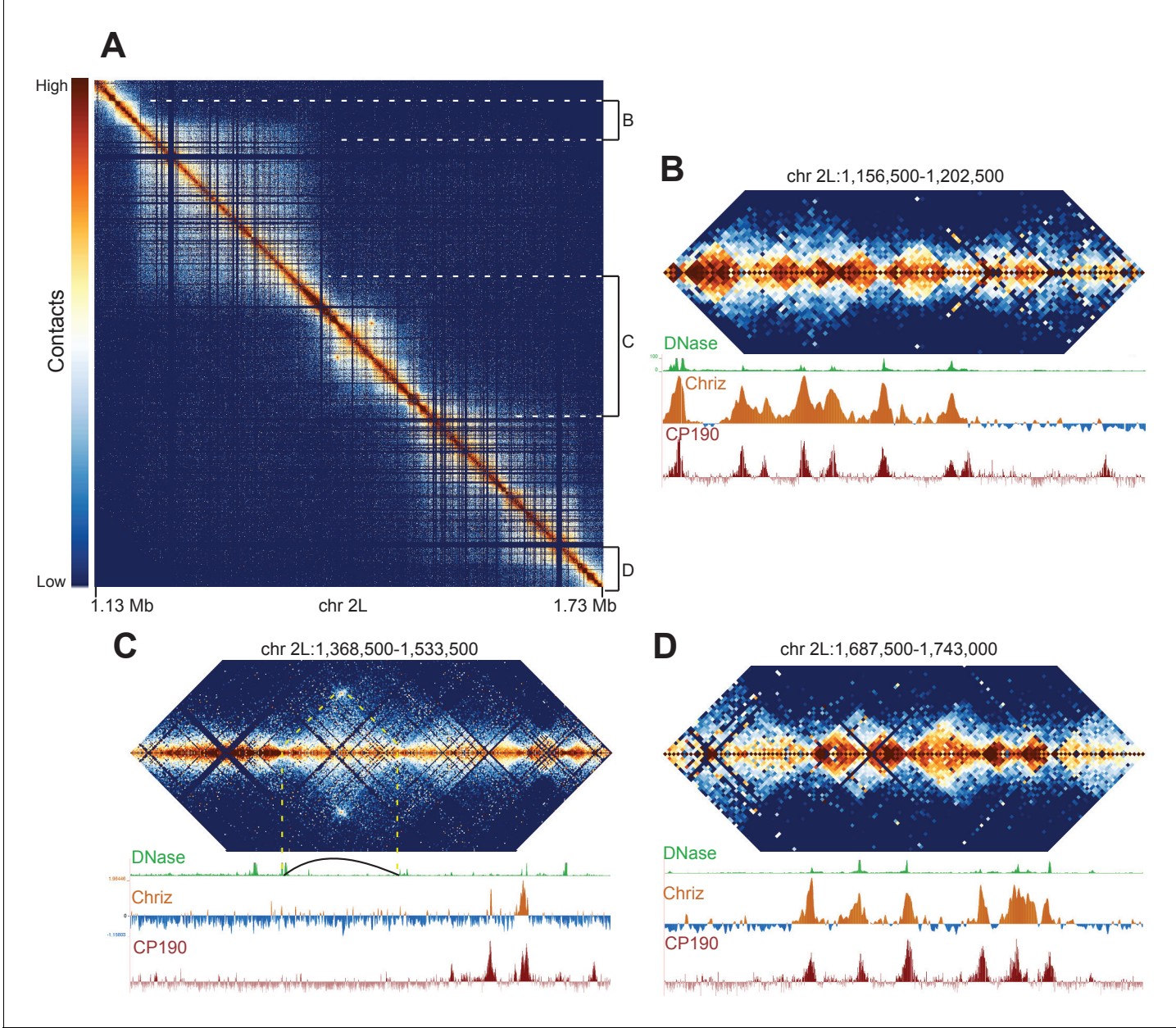

**Figure 6.** Complex topological structure of a region of chromosome 2L. Hi-C maps using 500 bp bins of the region of chromosome 2L corresponding to polytene band 22A1-2. This regions was shown by Eagen et al. to comprise a single TAD in Hi-C data viewed at 15 kb resolution, and is occasionally observed to contain an interband in polytene spreads. (**A**) View of the entire region, revealing complex internal structure. (**B–D**) Zoomed-in views of three regions comprising the left border (**B**), complex middle section (**C**), and right border of the larger region corresponding to the band/TAD investigated by Eagen et al., with associated stage 5 DNase accessibility, CP190, and Chriz (kc167 cells) profiles. Coordinates for this region are identical in dm3 and dm6.

DOI: https://doi.org/10.7554/eLife.29550.019

the left and right, divided by a series of smaller domains in the center. We suspect that this middle region accounts for the interband in Bridge's map, in a manner similar to the 10A1-1/10B1-2 region: a complex region consisting of several minor bands bounded by decompacted boundary regions appears as a single interband region under optical microscopy.

This region provides examples of a number of interesting features that we observe in our Hi-C data. First, the large TADs are bounded on both sides by gene-rich regions consisting of a number

of smaller topological domains (*Figure 6B,D*). The boundaries of large and small domains in this region nearly all share the common features of boundary elements: DNase hypersensitivity and binding of diagnostic insulator (e.g. CP190) and interband (CHRIZ) proteins. This region also contains a prominent example of an exception to this pattern: a loop is formed that appears to generate boundaries not associated with these characteristic protein binding events (*Figure 6C*, indicated by dotted yellow lines and loop). This example highlights a critical point: while the description we provide of the association between TAD boundaries, insulator elements, and decompacted interbands appears to describe the overwhelming majority of cases, there are counter-examples. Indeed given the extraordinary capacity of nature to innovate with respect to gene regulation and structures, we expect that animal genomes will provide no shortage novel chromosome topological and structural features for future investigations.

## Topological boundaries are nearly identical in anterior and posterior sections of the embryo

We next asked whether the boundaries we identified as boundary elements represent constitutive features of chromatin organization or whether their function might be regulated in a cell-type specific or developmental manner. We reasoned that, since different sets of patterning genes are transcribed in the anterior and posterior portions of the pre-gastrula *D. melanogaster* embryos, a comparison of chromatin interaction maps between anterior and posterior regions would reveal whether context, especially transcriptional state, affects the TAD/boundary structure of the genome. To this end, we performed two separate biological replicates of an experiment in which we sectioned several hundred mid stage 5 embryos along the anteroposterior midline, and produced deep-sequenced Hi-C libraries from the anterior and posterior halves in parallel.

Resulting Hi-C signals at boundaries are virtually identical in the two halves, despite substantially different gene expression profiles in these two embryonic regions (*Figure 7A*). Indeed, overall Hi-C signals are remarkably similar, with anterior and posterior samples correlating as strongly as replicates. Examination of individual loci at high resolution reveal consistent profiles and boundaries, notably including genes expressed differentially in the anterior or posterior (*Figure 7B*).

The correspondence of insulator boundary elements and interbands, and the chromatin extension model, implies that the chromatin accessibility of insulator regions will be a useful proxy for their functionality in structurally organizing the genome. Intriguingly, (*Van Bortle et al., 2014*) found that DNase accessibility of insulator protein-bound regions tracked with the ability of these sequences to block enhancer-promoter interactions in a cell-culture assay. We again sectioned embryos into anterior and posterior halves and performed ATAC-seq (*Buenrostro et al., 2013*) on pools of 20 embryo halves. ATAC-seq is a technique in which intact chromatin is treated with Tn5 transposase loaded with designed DNA sequences which are preferentially inserted into open, accessible chromatin regions. These insertions can be used to generate high-throughput sequencing libraries, producing data that is largely analogous to DNase-seq data.

Analysis of ATAC-seq signal at insulator boundary elements in anterior and posterior halves showed that these elements have nearly identical accessibility in these two samples (*Figure 7C*). Additionally, DNase-seq data from later embryonic stages that feature substantial tissue differentiation, transcription, and chromatin changes show highly consistent profiles at boundaries (*Figure 7C*, *Figure 7—figure supplement 1*). It is also striking that we observe significant enrichment of insulator proteins and Chriz at boundaries, despite the fact that boundaries were identified from Hi-C data from carefully-staged nc14 embryos (2–3 hr), whereas these ChIP datasets are derived from 0 to 12 hr old embryos or late embryonic cultured cells (Chriz). Together, these results are consistent with a model in which insulator-mediated chromatin organization is a constitutive feature of interphase chromatin of *D. melanogaster* embryos.

## Distal chromatin contacts in the early fly embryo

Many models of insulator function invoke physical contact between insulators to form 'looped' chromatin domains (*Fujioka et al., 2009*; *Yang and Corces, 2012*; *Kyrchanova and Georgiev, 2014*; *Kravchenko et al., 2005*), and a substantial literature exists demonstrating that many insulator proteins are able to interact with each other and to self-associate (*Büchner et al., 2000*; *Gause et al., 2001*; *Ghosh et al., 2001*; *Blanton et al., 2003*; *Pai et al., 2004*; *Mohan et al., 2007*;

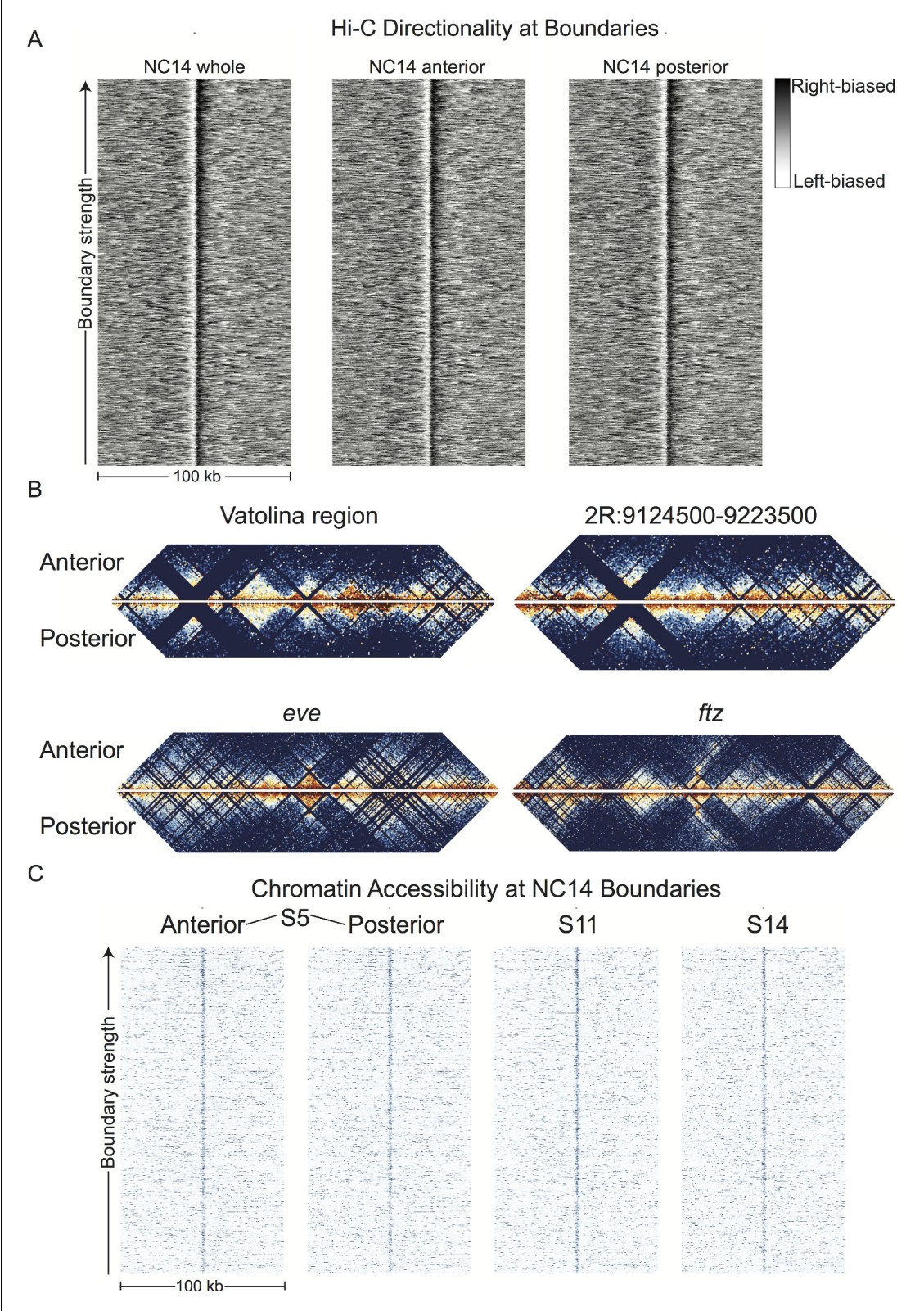

**Figure 7.** Hi-C signals from anterior and posterior halves of stage 5 embryos reveal highly similar chromatin topologies. (**A**) The distribution of Hi-C directionality scores in whole embryos, anterior, and posterior halves is shown around 952 topological boundaries identified jointly by computational and manual curation. (**B**) Heat maps of Hi-C data at 500 bp resolution at four example regions in anterior and posterior embryo halves. Plots represent the aggregate data of two biological and technical replicates each for anterior and posterior samples, and were prepared as in *Figure 2*. The regions

*Figure 7 continued*

shown are the region mapped by Vatolina et al. (dm6: X:11077500–11181000, dm3: X: 10971500–11075000), the example region from *Figure 2—figure supplement 4* (dm6: 2R:9124500–9223500, dm3: 2R:5012000–5111000)), and the genomic regions surrounding the *eve* (dm6: 2R:9903060–10056959, dm3: 2R:5790565–5944464) and *ftz* (dm6: 3R:6769234–6961333, dm3: 3R:2594956–2787055) loci. (C) Chromatin accessibility around topological boundaries as measured by ATAC-seq in anterior and posterior nc14 (S5) embryos and by DNase-seq on stage 11 and 14 embryos (X.-Y. *Li et al., 2011*).

DOI: https://doi.org/10.7554/eLife.29550.020

The following figure supplement is available for figure 7:

**Figure supplement 1.** Developmental time series of DNase accessibility at TAD boundaries.
DOI: https://doi.org/10.7554/eLife.29550.021

*Golovnin et al., 2007*; *Vogelmann et al., 2014*). In general, we do not observe looping interactions between domain boundaries in our Hi-C data. However, during manual calling of topological boundaries for the entire genome, we noted 46 prominent examples of interactions between non-adjacent domains (*Figure 8* and *Figure 8—figure supplements 1–10*, *Supplementary file 1*), in addition to the previously noted clustering of PcG-regulated Hox gene clusters (*Sexton et al., 2012*). Because the interactions we observed were not of a uniform character, we did not attempt to computationally search for all such phenomena in our data, nor do we claim that this list is necessarily complete. It is merely the union of two sets of 'interesting' loci identified in two independent rounds of visual inspection Hi-C maps for the entire genome, and we feel it is informative with respect to the nature and significance of distal interaction in the fly embryo.

The most visually striking locus, which we emphasize was identified in an unbiased manner without knowing its identify, is the locus containing the *Scr*, *ftz*, and *Antp* genes (*Figure 8A*). This locus has been extensively studied, and a number of regulatory elements have been identified that reside between the *ftz* and *Antp* genes but 'skip' the *ftz* promoter to regulate *Scr* (*Calhoun et al., 2002*; *Calhoun and Levine, 2003*). Consistent with this, we observe enriched contacts between the region containing the *Scr* promoter and a domain on the other side of *ftz* that contains the known *Scr*-targeting cis regulatory elements, while the *ftz*-containing domain makes minimal contact with its neighboring domains. Critically, we observe hot spots of apparent interaction between two sets of boundary elements (*Figure 8A*: 1 and 4, 2 and 3), suggesting that physical association of boundary elements (or their associated proteins) may play a role in this interaction.

Curiously, we detected a similar situation on the other side of *Scr*, where a domain containing the hox gene *Dfd* is 'skipped' over by the *Ama* locus to interact with a short element 3' of the *Scr* transcription unit (*Figure 8—figure supplement 1*). We also observe a similar arrangement near the *eve* locus (*Figure 8—figure supplement 2*). In these cases, a plausible topology is that the skipped domain is 'looped out', preventing interaction with neighbors, while the adjacent domains are brought into proximity.

In addition to these domain-skipping events, we observe a small number of looping interactions, where two distal loci show high levels of interaction, without the associated enriched interactions between the domains flanking the loop. In every case we observe, the loop forms between two domain boundaries. As shown in *Figure 8B*, one of these loops brings together the promoters of *knirps* and the related *knrl* (*knirps*-like) genes. Other loops connect the *achaete* and *scute* genes (*Figure 8—figure supplement 3*), *sloppy paired 1* and *sloppy paired 2* (*Figure 8—figure supplement 4*), and the promoter of *Ultrabithorax* with an element in its first intron (*Figure 8—figure supplement 5*).

These loci demonstrate that looping and domain-skipping events can be detected in our Hi-C data, but it appears that such interactions are rare and that looping does not occur between the overwhelming majority of insulator boundary elements. Nevertheless, it is striking that of the limited number of distal interactions we observed, many of them involve genes that are transcriptionally active during stage 5 of embryogenesis. This raises the possibility that these interactions may be stage or tissue-specific regulatory phenomena, and that more may be present in other tissues, developmental time points, or conditions.

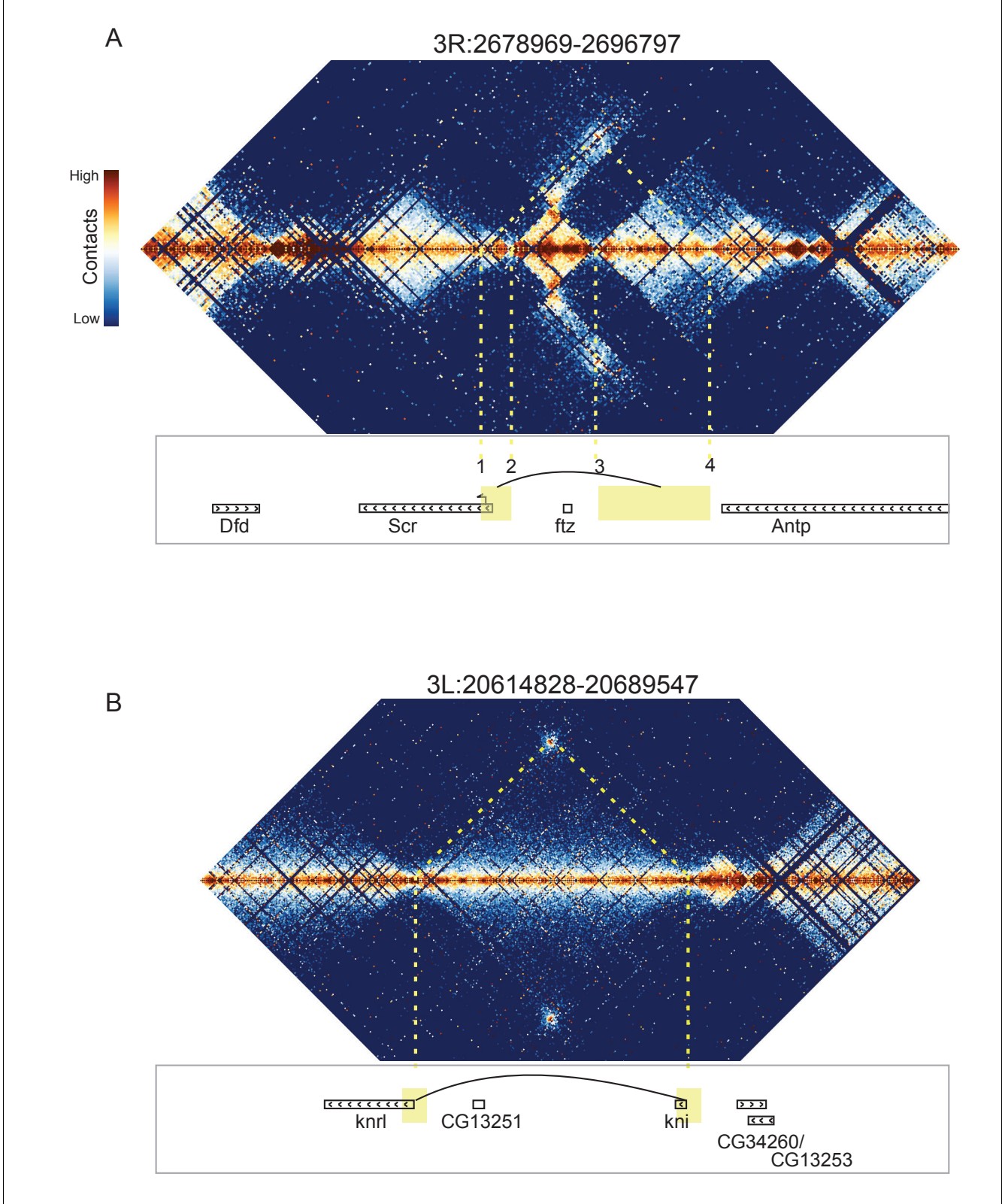

**Figure 8.** Looping and domain-skipping activity observed in nc14 chromatin. (**A**) An example of domain-skipping and looping at the *Scr-ftz-Antp* locus. *ftz* is contained within a domain that shows enriched Hi-C interactions between its boundaries, indicative of the formation of a looped domain. Adjacent domains show depleted interaction with the *ftz* domain and enriched interaction with each other, with especially strong contacts between the region containing the *Scr* promoter and characterized *Scr* regulatory elements 3' of the *Antp* locus (*Calhoun et al., 2002*; *Calhoun and Levine, 2003*). *Figure 8 continued on next page*

*Figure 8 continued*

Dotted lines connect features in the Hi-C map to the genomic locations of genes in this region. (B) A strong looping interaction between the *kni* locus and the 5' end of the related *knrl* (*kni*-like) gene. *kni* and *knrl* are known to have identical expression patterns and partially redundant, though distinct, domains of biochemical activity (*González-Gaitán et al., 1994*).

DOI: https://doi.org/10.7554/eLife.29550.022

The following figure supplements are available for figure 8:

**Figure supplement 1.** Distal chromatin contacts in stage 55 embryos.

DOI: https://doi.org/10.7554/eLife.29550.023

**Figure supplement 2.** Hi-C map for the locus spanning 2R:9973000-9988500.

DOI: https://doi.org/10.7554/eLife.29550.024

**Figure supplement 3.** Hi-C map for the locus spanning X:369500-396000.

DOI: https://doi.org/10.7554/eLife.29550.025

**Figure supplement 4.** Hi-C map for the locus spanning 2L:3825500-3837000.

DOI: https://doi.org/10.7554/eLife.29550.026

**Figure supplement 5.** Hi-C map for the locus spanning 3R:16720000-16730500.

DOI: https://doi.org/10.7554/eLife.29550.027

**Figure supplement 6.** Hi-C map for the locus spanning 3L:9003500-9040500.

DOI: https://doi.org/10.7554/eLife.29550.028

**Figure supplement 7.** Hi-C map for the locus spanning 3L:18186500-18234000.

DOI: https://doi.org/10.7554/eLife.29550.029

**Figure supplement 8.** Hi-C map for the locus spanning 2R:11474500-11528500.

DOI: https://doi.org/10.7554/eLife.29550.030

**Figure supplement 9.** Hi-C map for the locus spanning 3R:6999000-7038000.

DOI: https://doi.org/10.7554/eLife.29550.031

**Figure supplement 10.** Hi-C map for the locus spanning 3L:1367500-1464000.

DOI: https://doi.org/10.7554/eLife.29550.032

## Discussion

Several Hi-C studies in flies have identified enrichments of insulator proteins at TAD boundaries (*Sexton et al., 2012*; *Ulianov et al., 2016*; *Eagen et al., 2015*; *Mourad and Cuvier, 2016*) These studies varied in their resolution (due to use of 4- vs. 6-cutter enzymes and sequencing depth), methods (solution vs. in situ Hi-C), and, critically, in the methods used to identify TAD boundaries. As a result, each study relied on distinct sets of boundaries for analyses of the molecular features of these structures. We explored several methods to identify topological domains and associated boundaries and found that no single approach was sufficient to exhaustively identify all of these features in the genome. Rather, by using a combination of visual inspection of Hi-C maps at a large number of loci, unbiased hand-calling, and computational searches, we consistently observed a very close, two-way association between sites of combinatorial insulator protein binding (insulators) and the boundaries between topological domains. This result supports prior studies which found enriched insulator protein binding at topological boundaries, and extends this finding by localizing boundaries to discrete insulator elements. Hi-C data are exceptionally complex and reveal many layers of genomic organization, and we suspect that many questions in this field will only be resolved by the combined work of multiple groups using distinct analysis strategies and techniques.

Our most intriguing finding is the association of TAD boundaries with polytene interbands. The implication that these elements are decompacted, extended chromatin regions provides an attractive model in which simple physical separation explains multiple activities associated with insulators, including the ability to block enhancer-promoter interactions, prevent the spread of silenced chromatin, and organize chromatin structure.

A number of prior observations are consistent with the identity of insulators/boundaries as interbands. First, estimates suggest that there are ~5000 interbands constituting 5% of genomic DNA, with an average length of 2 kb (*Zhimulev, 1996*; *Vatolina et al., 2011a*), numbers that are in line with our estimates of boundary element length and number. Second, interbands are associated with insulator proteins, with CP190 appearing to be a constitutive feature of all or nearly all interbands (*Pai et al., 2004*; *Gerasimova et al., 2007*), which is precisely what we observe for boundary elements. Third, interbands and boundary elements are highly sensitive to DNase digestion

(*Vatolina et al., 2011b*). Fourth, interbands have been shown to contain the promoters and 5' ends of genes (*Jamrich et al., 1977*; *Sass, 1982*; *Sass and Bautz, 1982a*, *1982b*; *Rykowski et al., 1988*), and we see a strong enrichment for promoters oriented to transcribe away from boundaries, which would place upstream regulatory elements within or near the interband. Finally, deletion of both iso-forms of BEAF-32, the second-most highly enriched insulator protein at boundary elements, results in polytene X chromosomes that exhibit loss of banding and are wider and shorter than wild type, consistent with a loss of decompacted BEAF-32-bound regions (*Roy et al., 2007*). It is possible that interbands in polytene chromosomes result from multiple underlying molecular phenomena, but we believe it is likely that decompacted insulator elements constitute a significant fraction of these structures.

While we and others have not observed frequent looping of insulators in Hi-C data from fly tissue, our model of chromatin compaction at insulators is not mutually exclusive with a role for looping in the function of some insulators. Indeed, we have observed a limited set of cases in which interactions between boundaries seem to organize special genome structures with, at least in the case of the *Scr* locus, clear functional implications. It is likely that additional boundary-associated distal interactions will be found in other tissues and stages of fly development. However, we emphasize that these interactions are rare and do not appear to be general features of the function of boundary elements.

## Conclusions

The data presented here offer a picture of the structure of the interphase chromatin of *Drosophila* that attempts to unify years of studies of polytene chromosomes with modern genomic methods (*Figure 9*). In this picture, interphase chromatin consists of alternating stretches of compacted, folded chromatin domains separated by regions of decompacted, stretched regions. The compacted regions vary in size from a few to hundreds of kilobases and correspond to both polytene band regions and TADs in Hi-C data. Decompacted regions that separate these domains are short DNA elements that are defined by the strong binding of insulator proteins and correspond to polytene

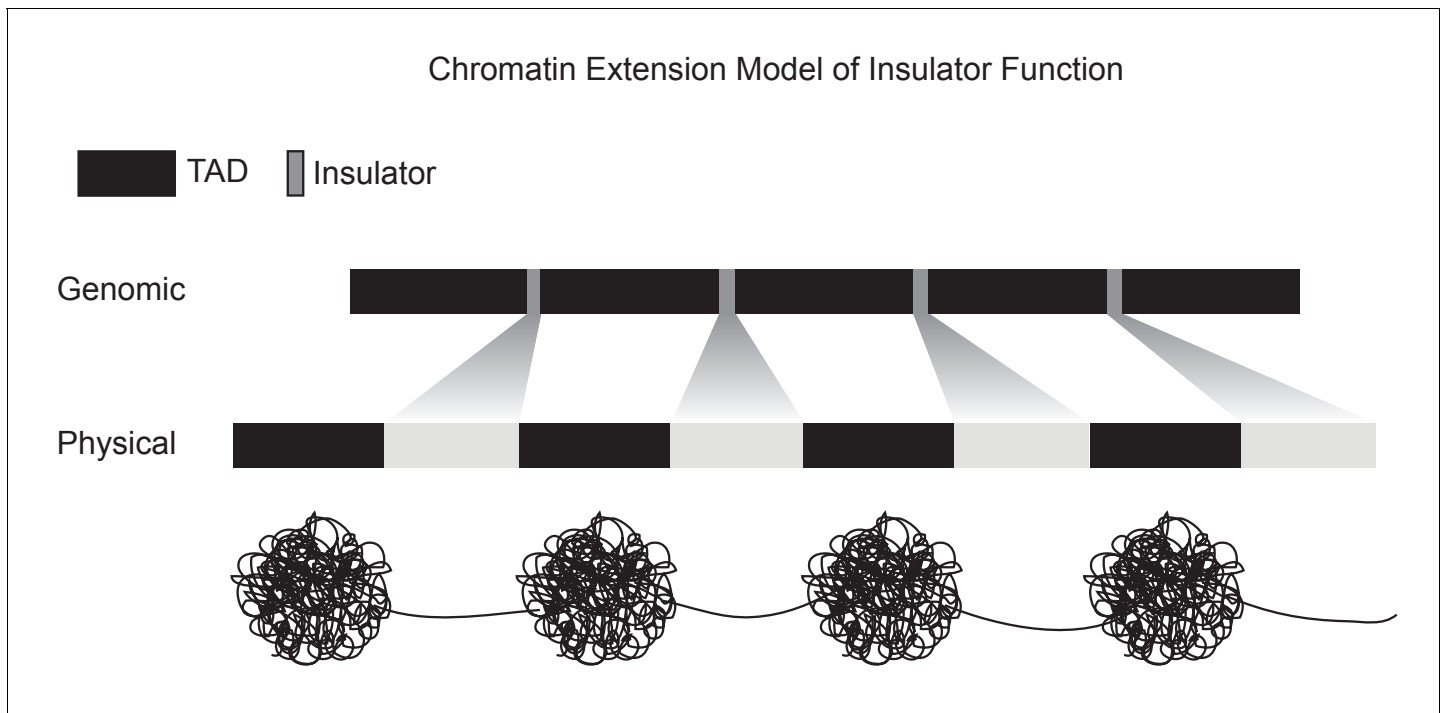

**Figure 9.** A chromatin extension model of insulator function. We propose a model in which insulators achieve domain separation by lowering the compaction ratio of bound chromatin, thereby converting the short lengths of insulator DNA (measured in base pairs) into large relative physical distances. By increasing the distance between domains, this model plausibly explains how insulators can achieve their diverse effects, including organizing chromatin structure, blocking enhancer-promoter interactions, and limiting the spread of chromatin silencing states.
DOI: https://doi.org/10.7554/eLife.29550.033

interbands and TAD boundaries (insulators). An intuitive view of this structure in a non-polytene context might resemble the well-worn 'beads on a string', in which insulator/interband regions are the string and bands/TADs form beads of various sizes. Future work, including experimental manipulation of the sequences underlying these structures, will focus on validating and refining this model, exploring how it fits into hierarchical levels of genome organization, and understanding its implications for genome function.

## Materials and methods

### Embryo collection, sorting, and sectioning

OregonR strain *D. melanogaster (RRID:FlyBase_FBst1000080)* embryos were collected on molasses plates seeded with fresh yeast paste from a population cage and aged to appropriate developmental stages, all at 25°C. Embryos were washed into nitex meshes, dechorionated by treatment with dilute bleach for 2 min, dipped briefly (15–20 s) in isopropanol, and gently rocked in fixative solution of (76.5% hexanes, 5% formaldehyde in 1x PBS) for 28–30 min. Embryos were then thoroughly washed in PBS with 0.5% triton and stored for no more than fivethree days at 4°C. For sample HiC-2/4, embryos were inspected under a light microscope to confirm that the vast majority corresponded to early cellularized blastoderm, and approximately 4000 embryos were used in the Hi-C protocol. For samples HiC-10, 12, 13–16, fixed embryos were hand-sorted under a light microscope as described in (*Harrison et al., 2011*), using morphological markers to identify early cellularized embryos (nc14, stage 5). For whole embryo experiments, sorted embryos were placed directly into the Hi-C protocol, with no more than 3 days having elapsed since fixation.

For sectioned embryos, hand-sorted embryos of precise developmental stages were first arranged in rows on a block of 1% agarose with bromophenol blue in a shared anterior-posterior orientation, with between 20–40 embryos per block. Aligned embryos were then transferred to the bottom of a plastic embedding mold (Sigma Aldrich E6032), the bottom of which had previously been coated with hexane glue, carefully keeping track of the anterior-posterior orientation of embryos by marking the cup with marker. Embryos were covered with clear frozen section compound (VWR 95057–838) and frozen at −80°C for up to two months. Frozen blocks wer4)e retrieved from the freezer and embryos rapidly sliced at approximately the mid-point by hand using a standard razor blade under a dissecting microscope. Anterior and posterior halves were separately transferred to microcentrifuge tubes containing ~200 μL PBS with 0.5% triton using an embryo pick (a tool of mysterious provenance that appears to be a clay sculpting tool). Successful transfer was confirmed visually by the presence of blue embryos which had absorbed bromophenol blue from the agarose block. Between transferring anterior and posterior halves, the pick was washed thoroughly with water and ethanol, and rubbed vigorously with kimwipes. We note that anterior and posterior half samples are precisely matched: samples HiC-13 and 14 contain the anterior and posterior halves (respectively) of the same embryos, and the same is true for HiC-15 and 16.

### Hi-C

Experimental procedure

Hi-C experiments were conducted as described in Rao (*Rao et al., 2014*), with slight modifications. For completeness, we describe the detailed protocol: Embryos (or halves) were suspended in 1X NEB2 buffer (NEB B7002) and homogenized on ice by douncing for several minutes each with the loose and tight dounces. Insoluble material (including nuclei) was pelleted by spinning for 5 min at 4500 x g in microcentrifuge cooled to 4°C (all wash steps used these conditions for pelleting). Nuclei were washed twice with 500 μL of 1x NEB2 buffer and then suspended in 125 μL of the same. 42.5 μL of 2% SDS was added and tubes placed at 65°C for 10 min, then returned to ice, followed by addition of 275 μL of 1x NEB2 buffer and 22 μL of 20% Triton X-100, then incubated at room temperature for 5 min. Samples were digested overnight with 1500 units of MboI by shaking at 37°C. The next day, samples were washed twice with 1X NEB2, resuspended in 100 μL 1X NEB2, and 15 μL of fill-in mix (1.5 μL 10x NEB2, 0.4 μL each of 10 mM dATP, dGTP, dTTP, 9 μL 0.4 mM biotin-14-dCTP, 2.5 μL 5 U/μL Klenow (NEB M0210), 1 μL water) was added, followed by 1.5 hr at 37°C. Samples were then washed twice with 500 μL 1X ligation buffer (10X: 0.5 M Tris-HCl pH7.4, 0.1M MgCl2, 0.1M DTT), resuspended in 135 μL of the same, then supplemented with 250 μL of ligation

mix (25 µL 10x ligation buffer, 25 µL 10% Triton X-100, 2.6 µL 10 mg/ml BSA, 2.6 µL 100 mM ATP, 196 µL water) and 2000 units of T4 DNA ligase (NEB M0202T) and incubated for 2 hr (or overnight) at room temperature. An additional 2000 units of ligase were added, followed by another 2 hr at room temperature. Cross-link reversal was carried out by adding 50 µL of 20 mg/mL proteinase K and incubating overnight at 65°C. An additional 50 µL proteinase K was then added followed by a 2 hr 65°C incubation. 0.1 volumes of 3M NaCl and 2 µL of glycoblue (Thermo Fisher AM9515) were added, then samples were extracted once with one volume of phenol pH 7.9, once with phenol-chloroform pH7.9, then precipitated with 3 volumes of EtOH. Washed pellets were resuspended in 130 µL water and treated with 1 µL of RNase A for 15 min at 37°C. DNA was fragmented using the Covaris instrument (Covaris, Woburn, MA) with peak power 140.0, duty factor 10.0, cycles/burst 200 for 80 s. Samples are brought to 300 µL total volume with water.

75 µL of Dynabeads MyOne Streptavidin C1 beads (Thermo Fisher 65001) were washed twice with 400 µL of tween wash buffer (TWB) (2X binding buffer [BB]: 100 µL of 1M Tris-HCl pH8, 20 µL 0.5 M EDTA, 4 mL of 5M NaCl, 5.88 mL water; TWB: 5 ml 2X binding buffer, 50 µL 10% Tween, 4.95 µL water), resuspended in 300 µL 2X BB, then added to 300 µL DNA. Samples were rocked at room temperature for 15 min, then washed once with TWB, twice with 1X BB, reclaimed on magnetic stand and resuspended in 100 µL 1X T4 DNA ligase buffer. Samples were then supplemented with end-repair mix (78 µL water, 10 µL 10X T4 DNA ligase buffer with ATP, 2 µL 25 mM dNTPs, 1 µL 10 U/µL T4 PNK (Thermo Fisher EK0031), 2 µL 5 U/µL Klenow, 3 µL 3 U/µL T4 DNA polymerase (Thermo Fisher EP0061)), incubated 30 min at room temp, washed as before, washed once with 100 µL 1X NEB2, and resuspended in 90 µL 1X NEB2. dA overhangs were added by adding 2 µL 10 mM dATP and 1 µL Klenow exo minus (NEB M0212S), incubating at 37°C for 30 min. Beads were washed as before, washed once with 100 µL 1X Quickligase (NEB M2200S) buffer, resuspended in 50 µL 1X Quickligase buffer, then supplemented with 3 µL Illumina adaptors and 1 µL Quickligase. Samples were incubated 15 min at room temperature, then washed twice with TWB, twice with 1X BB, twice with 200 µL TLE, and resuspend in 50 µL TLE. Beads are stable at 4°C, but were always amplified quickly. 100 µL (or more) of phusion PCR reaction was prepared (50 µL 2X Phusion master mix, 1 µL 100 µM forward primer [5-AATGATACGGCGACCACCGAG-3], 1 µL 100 µM reverse primer [5-CAAGCAGAAGACGGCATACGAG-3], 10 µL of beads with Hi-C library attached, 38 µL water). Reaction was mixed well and split into separate 12 µL reactions. Thermocycler conditions were 16 cycles of 98°C for 30 s, 63°C for 30 s, 72°C for 2 m. Reactions were pooled and loaded on a 2% agarose gel. Fragments corresponding to an insert size of ~300 bp (amplicon size of 421 bp) were excised from the gel, purified with the Zymo Gel DNA Recovery Kit (D4001T, Zymo), and submitted for sequencing at the Vincent J. Coates Genomic Sequencing Laboratory (Berkeley, CA).

## Read processing and mapping

Our analysis routine was adapted by examining the approaches of multiple groups (*Lieberman-Aiden et al., 2009*; *Sexton et al., 2012*; *Rao et al., 2014*; *Crane et al., 2015*) in addition to procedures we developed independently. All analysis was performed with custom Python, R, and Perl scripts (*Stadler, 2017*; copy archived at https://github.com/elifesciences-publications/Stadlerlab-hi-c) except where noted. Single-ends of demultiplexed reads were separately mapped using Bowtie (*Langmead et al., 2009*) (parameters: -m1 –best –strata) to the *D. melanogaster* genome dm6 (R6.17). Due to the formation of chimeric reads intrinsic to the Hi-C procedure, reads can fail to properly map if the ligation junction lies within the 100 bp read. To address this, we used an iterative mapping procedure, in which we began by mapping the first 20 nt of the reads (using Bowtie's –trim3 feature). Unique mappings were kept, reads that failed to map were stored, and the procedure was repeated on the multiply-mapping reads, incrementing the length of sequence to map by 7 nt each round (attempt to uniquely map using first 20, first 27, first 34...). We found that this method gave 5–10% increases in yield of mapped reads over a procedure in which we attempted to explicitly detect and trim ligation junctions from reads. Uniquely mapping reads from all iterations were collated as a single file.

Uniquely-mapping single-ends were paired based on read identity, and only pairs with two uniquely-mapping ends were retained. Duplicate reads that shared identical 5' mapping positions were removed. Resulting paired, collapsed, uniquely mapping reads were then inspected for quality. Primary indicators of successful Hi-C libraries were the distance distribution of mapped pairs and the

relative frequencies of reads in the four orientations described by *Rao et al., 2014*, in-in, in-out, out-in, and out-out. In all of our libraries, we detect some ~3–15% reads that appear to be simple genomic sequence, not the result of a Hi-C ligation event. These reads are readily detected by examining the size distributions of in-out reads (the orientation expected from standard genomic sequence) compared with the other three orientations. The in-out reads have a unique hump of reads showing a distance distribution of ~150–500 bp, varying slightly from sample to sample. In-out reads pairs spanning less than 500 bp were removed from further analysis.

## Computational topological boundary detection

We explored a number of ways of identifying boundaries from directionality data. In the end, the most robust was to use a simple heuristic that at a boundary, by definition, regions to the left show left-bias and regions to the right show right bias. While attempts to derive a boundary score from a comparison of directionality scores upstream and downstream showed susceptibility to noise and artifacts, requiring expected upstream and downstream behavior allowed robust detection of sets of boundary elements. We describe the complete procedure below.

Read counts were assigned to 500 bp bins for all genomic bin combinations within 500 kb of the diagonal. Local directionality scores were calculated for each bin by summing the counts linking the bin to regions in a window encompassing the genomic regions between 1 and 15 kb from the bin (skipping the two proximal 500 bp bins, summing the next 28) upstream and downstream, then taking the log (10) ratio of downstream to upstream. These parameters were determined by visually comparing local directionality scores from a range of inputs to Hi-C heat maps for a number of genomic regions, identifying parameters in which directionality transitions reflected boundaries evident in the heat maps. We observed high levels of noise in the directionality metric in regions of low read coverage. To suppress these noisy signals, we devised a weighted local directionality score to weight these scores based on the total number of reads used in the calculation. We experimented with a variety of scaling factors a such that w = [read count]a and found that a weighting of a = 0.5 worked well to reduce signal from low-read regions. From these directional scores, sites were first selected for which the mean directionality score of the 5 adjacent upstream bins was less than $-2$, and the mean for the 5 adjacent downstream bins was greater than 2. Boundary scores were assigned to resulting bins by subtracting the sum of the directionality scores for the 5 adjacent upstream bins from the 5 adjacent downstream bins. An issue with this scoring system is that bins that lack MboI sites can cause inflated directionality scores in adjacent regions. To address this, we simply assigned a boundary score of 0 to any bin with more than one such bin in its radius. The resulting distribution of boundary scores is dominated by series of consecutive bins with large boundary score maximums, which is uninformative since these scores are essentially derived from the same data (window shifted by one bin). We therefore merged adjacent bins that passed the cutoff and selected only the bin with the maximum boundary score within a contiguous block. By sorting the resulting table on the boundary score, we were able to select sets of candidate boundaries of various strengths for analysis.

In additional to these computationally-identified boundary locations, we manually called boundaries for the entire genome. An R script serially displayed Hi-C heat maps of 250 kb genomic windows and recorded the genomic coordinates of mouse clicks made at visually-identified boundaries. The human caller was unaware of any features of the regions examined other than the Hi-C maps, and was unaware of the locations being displayed in a given plot.

## Sequence analysis

We used simple custom Python scripts to count the occurrences of all words of length 4, 5, 6 and 7 in 500 bp windows from 10,000 bp upstream to 10,000 bp downstream of the 500 bp window identified as a boundary. We then computed a simple enrichment score for each unique word equal to the counts of that word and its reverse complement in the boundary divided by the mean counts for the word and its reverse complement in the remaining windows. We noticed that many of the words identified as enriched in this analysis were also enriched in the 500 bp bins immediately flanking the boundary. We therefore updated our enrichment score for each word to be the mean of the counts of the word and its reverse complement in the boundary and the 500 bp bins immediately adjacent

to it (three bins in total) divided by the mean counts of the word and its reverse complement in the remaining 38 bins. Counts and scores for all words are provided in the supplemental materials.

## ATAC-seq

### Experimental procedure

Early nc14 embryos were placed in ATAC-seq lysis buffer (*Buenrostro et al., 2013*) without detergent, with 5% glycerol added. Embryos were then taken out of the freezing solution and placed onto a glass slide which was then put on dry ice for 2 min. Once embryos were completely frozen, the glass slide was removed and embryos were sliced with a razor blade chilled in dry ice. Once sliced embryo halves were moved to tubes containing ATAC-seq lysis buffer with 0.15 mM spermine added to help stabilize chromatin. Embryo halves were then homogenized using single use plastic pestles. IGEPal CA-630 was added to a final concentration of 0.1%. After a 10 min incubation nuclei were spun down and resuspended in water. Twenty halves were added to the transposition reaction containing 25 µl of 2x TD buffer (Illumina), and 2.5 ul of Tn5 enzyme (Illumina) and the reaction was incubated at 37°C for 30 min as in *Buenrostro et al. (2013)*. Transposed DNA was purified using Qiagen Minelute kit. Libraries were then amplified using phusion 2x master mix (NEB) and indexed primers from Illumina. Libraries were then purified with Ampure Beads and sequenced on the Hiseq4000 using 100 bp paired end reads.

### Analysis

Fastq files were aligned to the *Drosophila* Dm3 genome with Bowtie2 (*Langmead and Salzberg, 2012*) using the following parameters: −5 5–3 5 N 1 -X 2000 –local –very-sensitive-local. Sam files were then sorted and converted to Bam files using Samtools (*Li et al., 2009*), only keeping mapped, properly paired reads with a MAPq score of 30 or higher using -q 30. Bams were then converted to Bed files with bedtools and shifted using a custom shell script to reflect a 4 bp increase on the plus strand and a 5 bp decrease on the minus strand as recommended by *Buenrostro et al. (2013)*. Finally shifted bed files were converted into wig files using custom scripts and wig files which were uploaded to the genome browser. Wig files were normalized to reflect 10 million mapped reads.

## Sample size determination

No explicit statistical method was used to compute sample size. All unique experiments were prepared in duplicate.

## Acknowledgements

We are especially thankful to Emily Brown for her assistance in adapting Hi-C to fly embryos, to Xiao-Yong Li for help with embryo sorting and with optimizing the fixation and chromatin isolation protocols, and to Steven Kuntz for assistance with developing embryo sectioning protocols. We thank Mustafa Mir, Xavier Darzacq and members of the Eisen and Darzacq labs for critical discussions and advice supplied throughout the work. MS was supported by an American Cancer Society postdoctoral fellowship (126730-PF-14-256-01-DDC), JH was supported by the National Science Foundation Graduate Research Fellows Program, and the work was supported by an HHMI investigator award to ME.

## Additional information

### Funding

| Funder | Grant reference number | Author |
| --- | --- | --- |
| Howard Hughes Medical Institute | | Michael Eisen |
| American Cancer Society | 126730-PF-14-256-01-DDC | Michael R Stadler |
| National Science Foundation | | Jenna E Haines |

The funders had no role in study design, data collection and interpretation, or the decision to submit the work for publication.

## Author contributions
Michael R Stadler, Conceptualization, Data curation, Software, Formal analysis, Investigation, Visualization, Methodology, Writing—original draft, Writing—review and editing; Jenna E Haines, Software, Formal analysis, Investigation, Methodology, Writing—review and editing; Michael B Eisen, Conceptualization, Software, Formal analysis, Supervision, Funding acquisition, Methodology, Writing—review and editing

## Author ORCIDs
Michael R Stadler ![ORCID] http://orcid.org/0000-0002-3333-4184
Michael B Eisen ![ORCID] https://orcid.org/0000-0002-7528-738X

## Decision letter and Author response
Decision letter https://doi.org/10.7554/eLife.29550.040
Author response https://doi.org/10.7554/eLife.29550.041

## Additional files

### Supplementary files
• Supplementary file 1. Supplemental tables. S1: List of samples used to generate Hi-C libraries. S2: Sequencing and processing statistics for Hi-C libraries. S3: Location of loops and domain-skipping identified in nc14 embryos. S4: Manually called boundaries. S5: Computationally identified boundaries. S6: Representative boundary set, merge of manual and computational curations.
DOI: https://doi.org/10.7554/eLife.29550.034

• Supplementary file 2. Source data for ChIP and similar enrichment files, e.g. in *Figure 3*.
DOI: https://doi.org/10.7554/eLife.29550.035

• Supplementary file 3. Source data for Hi-C heatmaps.
DOI: https://doi.org/10.7554/eLife.29550.036

### Major datasets
The following dataset was generated:

| Author(s) | Year | Dataset title | Dataset URL | Database, license, and accessibility information |
|---|---|---|---|---|
| Stadler MR, Haines JE, Eisen MB | 2017 | Hi-C of early Drosophila melanogaster embryos | https://www.ncbi.nlm.nih.gov/geo/query/acc.cgi?acc=GSE100370 | Publicly available at NCBI Gene Expression Omnibus (accession no: GSE100370) |

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
