## [Decision Letter]

Thank you for submitting your article "Convergence of topological domain boundaries, insulators, and polytene interbands in the early *Drosophila* embryo" for consideration by *eLife*. Your article has been favorably evaluated by Jessica Tyler (Senior Editor) and three reviewers, one of whom is a member of our Board of Reviewing Editors. The reviewers have opted to remain anonymous.

The reviewers have discussed the reviews with one another and the Reviewing Editor has drafted this decision to help you prepare a revised submission.

Summary:

Stadler et al. present a HiC analysis of *Drosophila* chromatin from nuclear cycle 12 and cycle 14 embryos in order to address a variety of questions, with an emphasis on how insulators are related to chromatin topological domains (TADs). They conclude that TAD boundary regions are enriched for insulator binding sites, contain promoters, and are marked by accessible chromatin. The TAD boundary regions are found to be constant despite differences in gene expression between the anterior and posterior of the embryo, and they are detectable prior to the onset of zygotic gene expression. By comparing TAD boundaries to sites of insulator protein binding, and to very limited available information on polytene interbands, the authors propose a model of insulator function and suggest that their work "unifies years of polytene chromosome research.

Essential revisions:

Unfortunately, there are significant problems with the supporting data, requiring major revisions. The datasets upon which these conclusions are based (19 or 88 million contacts) appear to be significantly smaller than in recent publications (Hug et al.2017: – 400 million contacts; or even in older studies Corces 2012- 120 million contacts.) Barring some misunderstanding, neither 19 million or 88 million contacts would be adequate to achieve the claimed resolution of less than 1 kb. The questions addressed in the manuscript depend on achieving a high level of resolution, consequently, for this paper to be considered further, the authors need to address this problem. Additional sequencing of their libraries or combination of data with published datasets are possibilities to consider. If this problem can be resolved, and the resulting higher resolution analysis still supports the correlation between TAD boundaries and interbands, then a revised version of the paper will be considered for publication.

In a revised version, the claimed correspondence between TAD boundaries and polytene interbands needs to be solidified. The possible confounding effect of zones of active transcription should be addressed. They should acknowledge the highly tentative nature of a theory that has been tested at the very most in 5 out of > 5,000 cases.

*Reviewer #1:*

First, the claim that the data achieves a resolution of 1kb or less needs to be better justified. If the authors' HiC data is so accurate, why were only 947 high confidence junctions defined? Since the most robust domain junctions are those associated with large bands, why did the authors not identify all the junctions of those bands? How could the very fine junctions between almost invisible bands in region 10A be defined, when so many domains in the genome as a whole were not?

Second, it is not clear from the data presented in Figure 2 and supplements, that one can robustly define regions whose function is to separate domains from simple uncertainty in the domain edges. The surprising uniformity in size (500-1000bp) of the authors elements might have been an artifact of the requirement for a major directionality change. The "insulator" proteins used in the definition of these junctions do not appear to be very specific, as the authors own and third party data shows that these proteins bind at multiple sites in the genome, most of which are not at domain boundaries. The paper should calculate the numerical enrichment of each of these proteins at domain junctions vs. other sites in the genome.

The data presented in cytogenetic region 10A-10B, which has been intensively studied in the Zhimulev laboratory, was quite interesting, although it falls a little short of the "perfect" correspondence claimed by the authors. In particular, in Figure 5, the junction between the heavy band 10A1-2 and 10A3 (a virtually invisible band in polytenes) was not actually detected in the hand or high confidence boundary calls, but was arbitrarily added to the figure as a dashed line so it would fit expectations. As a result, the next junction, the strongest in the region, is scored as the junction between 10A3 and 10A4, i.e. between an invisible band and a weak band. It seems more likely that this strong boundary is actually the end of 10A1-2, and that there are only 4 rather than 5 domains between the two major bands in this region. This would be consistent with the authors own detection of only 4 boundaries in their unbiased and in the high confidence datasets. The authors separate band 10A6 from 10A7 following Zhimulev, even though he states that this separation is almost never observed, nor did Bridges report separate bands in his revised map. There is no DNAase or convincing directionality data for this separation. Since HiC would average conformational differences between cells, the data should reflect the common configuration, not a rare configuration. The authors should clearly state in the text that they are not using their own boundary measurements but making a special interpretation specifically for this experiment, rather than hiding that admission at the end of the figure legend.

Finally, the authors should comment on the correspondence between the boundaries and polytene bands. It stands to reason that a method capable of resolving interbands would need to have superb resolution of bands. How well do the boundaries defined in this manuscript fit with previous cytological and HiC reports for major bands? It is an interesting question, not only for its relevance to the question of HiC reproducibility between labs, but because different tissue sources, including salivary glands, tissue culture cells and embryos have been used. Is it possible that different tissues have the same general organization into TADs, but the detailed boundaries of the TADs differ in some cases?

*Reviewer #2:*

The data largely are clear and compelling except for a few issues:

1) The ChIP data for insulator binding sites are from 0-12 hour embryos, a much broader developmental window than the two time points examined for the TAD mapping. Thus authors should discuss how this may or may not affect their conclusions. Given that the insulator ChIP data are not from the same restricted developmental time as the TAD maps, it would be useful to discuss how the boundary elements mapped here compare to sites of localization of CHRIZ (Chromator), even though they have been determined in S2 and Kc cells. Zhimulev et al. concluded that CHRIZ binding was the most diagnostic feature of interbands, so this could serve as further confirmation of the coincidence of boundary regions with interbands.

2) Please provide color scale bars for the heatmaps of contact probabilities. This will aid readers in comparing Figure 1 and 7. Please provide these also for Figure 3 and the other protein binding figures.

3) The looped regions shown in Figure 8 and the corresponding supplemental figures are difficult for readers unfamiliar with Hi-C to visualize. The interaction in Figure 8 is clear, but could the authors provide a clearer graphic to explain the loops in Figure 8? Could they also highlight the diagnostic loop signatures in the supplemental figures?

*Reviewer #3:*

The manuscript is interesting but many of the observations, including chromatin 3D organization in *Drosophila* embryos and association of boundaries with insulator proteins, have been published previously. The following are other more specific concerns:

1) Authors should explain how they calculate the resolution of Hi-C data. Because the frequency of interactions decays with distance, authors should keep in mind that resolution will depend on distance. If I understand Table S1 Sample Information in Supplementary file 1 correctly, the authors made 2 libraries from nc12 and 3 libraries from nc14. I used Juicer to process the data for these two libraries and the results are as follows:

Experiment description: nc12

Sequenced Read Pairs: 93,483,816

Normal Paired: 65,205,173 (69.75%)

Chimeric Paired: 16,627,312 (17.79%)

Chimeric Ambiguous: 6,549,875 (7.01%)

Unmapped: 5,101,456 (5.46%)

Ligation Motif Present: 45,319,979 (48.48%)

Alignable (Normal+Chimeric Paired): 81,832,485 (87.54%)

Unique Reads: 26,913,205 (28.79%)

PCR Duplicates: 54,884,068 (58.71%)

Optical Duplicates: 35,212 (0.04%)

Library Complexity Estimate: 28,537,160

Intra-fragment Reads: 760,888 (0.81% / 2.83%)

Below MAPQ Threshold: 6,649,224 (7.11% / 24.71%)

Hi-C Contacts: 19,503,093 (20.86% / 72.47%)

Ligation Motif Present: 7,811,696 (8.36% / 29.03%)

3' Bias (Long Range): 84% – 16%

Pair Type% (L-I-O-R): 25% – 25% – 25% – 25%

Inter-chromosomal: 1,545,226 (1.65% / 5.74%)

Intra-chromosomal: 17,957,867 (19.21% / 66.73%)

Short Range (<20Kb): 8,504,570 (9.10% / 31.60%)

Long Range (>20Kb): 9,451,838 (10.11% / 35.12%)

Experiment description: nc14

Sequenced Read Pairs: 393,928,165

Normal Paired: 144,374,310 (36.65%)

Chimeric Paired: 24,393,845 (6.19%)

Chimeric Ambiguous: 9,682,276 (2.46%)

Unmapped: 6,884,964 (1.75%)

Ligation Motif Present: 127,838,532 (32.45%)

Alignable (Normal+Chimeric Paired): 168,768,155 (42.84%)

Unique Reads: 119,180,726 (30.25%)

PCR Duplicates: 49,523,419 (12.57%)

Optical Duplicates: 64,010 (0.02%)

Library Complexity Estimate: 227,824,974

Intra-fragment Reads: 1,146,075 (0.29% / 0.96%)

Below MAPQ Threshold: 29,415,704 (7.47% / 24.68%)

Hi-C Contacts: 88,618,947 (22.50% / 74.36%)

Ligation Motif Present: 41,585,911 (10.56% / 34.89%)

3' Bias (Long Range): 83% – 17%

Pair Type% (L-I-O-R): 25% – 25% – 25% – 25%

Inter-chromosomal: 5,442,733 (1.38% / 4.57%)

Intra-chromosomal: 83,176,214 (21.11% / 69.79%)

Short Range (<20Kb): 49,153,740 (12.48% / 41.24%)

Long Range (>20Kb): 34,004,040 (8.63% / 28.53%)

This suggests that nc12 has 19 million HiC contacts and nc14 has 88.6 million HiC contacts. It is impossible to have a 500 bp resolution, as the authors claim, with these numbers of reads. The authors state that "We conclude that these Hi-C are of high quality and reproducibility". There are currently two HiC datasets from *Drosophila* available in Juicebox, and both of them have around 1 billion mapped reads. The HiC libraries made with *Drosophila* embryos by Hug et al. have around 400 million mapped reads. Therefore, the libraries presented in this study cannot be described as high quality compared to others already available. Furthermore, the nc12 library is of much lower quality than the nc14 library. This is a critical issue when interpreting the results.

2) Authors should use the dm6 genome instead of dm3.

3) It is strange that interbands in polytene chromosomes correspond to boundaries. These structures have always been thought to contain active genes. Hug et al. identify very small domain that they also claim are boundaries. In their Hi-C data (see for example Figure 1), once can observed interactions between these small domains that resemble compartmental interactions in mammalian Hi-C data, suggesting that these small domains do not correspond to boundaries, they correspond to active domains. These features are not observed in the Hi-C data presented in this manuscript, which may be due to the fewer reads in the Hi-C data. Authors need to consider this and try to differentiate between effects of transcription and effects of insulator proteins.

4) Authors cannot directly compare features of nc12 and nc14 embryos unless they have the same number of Hi-C contacts in both samples. I think it is critical to obtain more reads for the nc12 libraries.

---

## [Author Response]

Essential revisions:Unfortunately, there are significant problems with the supporting data, requiring major revisions. The datasets upon which these conclusions are based (19 or 88 million contacts) appear to be significantly smaller than in recent publications (Hug et al.2017: – 400 million contacts; or even in older studies Corces 2012- 120 million contacts.) Barring some misunderstanding, neither 19 million or 88 million contacts would be adequate to achieve the claimed resolution of less than 1 kb. The questions addressed in the manuscript depend on achieving a high level of resolution, consequently, for this paper to be considered further, the authors need to address this problem. Additional sequencing of their libraries or combination of data with published datasets are possibilities to consider. If this problem can be resolved, and the resulting higher resolution analysis still supports the correlation between TAD boundaries and interbands, then a revised version of the paper will be considered for publication.

There does appear to be some misunderstanding. Our original manuscript was based on 263,868,080 Hi-C linkages from nc14 embryos. A complete description of the read totals for individual datasets, including a full accounting of reads removed at various processing steps, was supplied in Table S2 Sequencing Statistics in Supplementary file 1. Further, the processing pipeline used was described in detail in the Materials and methods section.

We appreciate the interest and efforts of reviewer 3 to analyze our raw data. We are unsure how they arrived at the total of 88 million nc14 contacts, but have simplified the naming of the original raw data files to make their relationships clearer in hopes that future users of the data will be able to work with them optimally.

Despite this confusion, we felt it was prudent to generate additional sequencing data from these libraries, and performed additional sequencing of four of our nc14 libraries and now have a total of 451,978,433 Hi-C contacts from nc14 embryos. The overall character of the data is unchanged, and none of our original observations are altered. We believe this is sufficient to achieve sufficient resolution to support the conclusions of the manuscript.

In a revised version, the claimed correspondence between TAD boundaries and polytene interbands needs to be solidified. The possible confounding effect of zones of active transcription should be addressed. They should acknowledge the highly tentative nature of a theory that has been tested at the very most in 5 out of > 5,000 cases.

We agree and have added additional discussion of the possible effects of active transcription on TAD boundary structure. The fact that TAD boundaries are invariant between anterior and posterior embryo sections, show only partial correlation with RNA polymerase occupancy (compared to exceptionally strong correspondence with insulator protein binding), and that TAD structures are not significantly altered in the presence of transcriptional inhibitors (Hug et al.) argue that transcriptional activity is not a strong driver of TAD boundary formation. With respect to the band/interband pattern of the 10A-B region, only a subset of the promoters near the identified TAD boundaries appear to be transcribed in nc14 embryos, based on RNAPII occupancy and RNAseq. We have added additional commentary on the tentative nature of the TAD boundary-interband association and the limited availability of high-resolution polytene mapping data.

Reviewer #1:First, the claim that the data achieves a resolution of 1kb or less needs to be better justified. If the authors' HiC data is so accurate, why were only 947 high confidence junctions defined? Since the most robust domain junctions are those associated with large bands, why did the authors not identify all the junctions of those bands? How could the very fine junctions between almost invisible bands in region 10A be defined, when so many domains in the genome as a whole were not?

In retrospect we agree that claims about the resolution of the data are fraught, especially because there is no clear standard for what it means to assign a numerical value to HiC resolution. We have therefore removed these specific claims about resolution from the manuscript and focus instead on what the data actually show.

The list of 947 high confidence junctions was not meant to be a comprehensive list of junctions. This set was chosen as a representative set to analyze the features of boundaries, not as a comprehensive list, and is a very conservative list generated from the union of the highest-scoring computationally-identified boundaries and hand calls. We have attempted to make this distinction clear in the text.

To the broader issue of calling boundaries: as with all genomic datasets it is difficult to define an exact number of features of a certain class, since one can identify them with various algorithms and choices must be made about cutoffs, parameters, etc. with different justifiable choices leading to wildly different numbers of identifiable features with differing degrees of overlap. We identified a much larger set (~3,200) of hand-called boundaries. The number of computationally-identified boundaries varies depending on the thresholds used, but is likely over ~2,500. We have included the full list of hand-called and computationally-identified boundaries in supplemental data, as well as supplying discussion about the complexities of boundary calling procedures and estimates of the total number of boundaries present in the genome.

Second, it is not clear from the data presented in Figure 2 and supplements, that one can robustly define regions whose function is to separate domains from simple uncertainty in the domain edges. The surprising uniformity in size (500-1000bp) of the authors elements might have been an artifact of the requirement for a major directionality change. The "insulator" proteins used in the definition of these junctions do not appear to be very specific, as the authors own and third party data shows that these proteins bind at multiple sites in the genome, most of which are not at domain boundaries. The paper should calculate the numerical enrichment of each of these proteins at domain junctions vs. other sites in the genome.

We agree that there is uncertainty in our ability to define the precise location of individual boundaries, but believe that collectively the data strongly support our conclusions about the correspondence between these regions. We have, as suggested, calculated the numerical enrichment and have added this to the paper.

The data presented in cytogenetic region 10A-10B, which has been intensively studied in the Zhimulev laboratory, was quite interesting, although it falls a little short of the "perfect" correspondence claimed by the authors. In particular, in Figure 5, the junction between the heavy band 10A1-2 and 10A3 (a virtually invisible band in polytenes) was not actually detected in the hand or high confidence boundary calls, but was arbitrarily added to the figure as a dashed line so it would fit expectations. As a result, the next junction, the strongest in the region, is scored as the junction between 10A3 and 10A4, i.e. between an invisible band and a weak band. It seems more likely that this strong boundary is actually the end of 10A1-2, and that there are only 4 rather than 5 domains between the two major bands in this region. This would be consistent with the authors own detection of only 4 boundaries in their unbiased and in the high confidence datasets. The authors separate band 10A6 from 10A7 following Zhimulev, even though he states that this separation is almost never observed, nor did Bridges report separate bands in his revised map. There is no DNAase or convincing directionality data for this separation. Since HiC would average conformational differences between cells, the data should reflect the common configuration, not a rare configuration. The authors should clearly state in the text that they are not using their own boundary measurements but making a special interpretation specifically for this experiment, rather than hiding that admission at the end of the figure legend.

We thank the reviewer for his/her critical eye, and we agree that the correspondence between our Hi-C maps and Zhimulev’s banding analysis should be clarified. First, it is important to point out that the band/interband assignments presented in Figure 5, and their genomic locations, are assigned according to Vatolina et al., not from our measurements. As the reviewer points out, our data do not strongly support the existence of the minor band 10A-3, and combined with the fact that it is virtually invisible in polytenes, we agree that it is likely not real and have indicated this in the text.

The separation of 10A6 and 10A7 is also a fascinating case. Unfortunately, the exact junction between these domains is obscured by a lack of MboI cut sites, an unfortunate limitation of our Hi-C data. Nevertheless, we do seem to observe evidence for the existence of these bands (the stronger signal within them and weaker signal for contacts between them). Additionally, while there is not a strong signal of DNase digestion at this putative boundary, there is strong binding of CP190, BEAF-32, and dCTCF. However, the data also clearly show significant interactions between these two putative domains (the light orange regions near the top of the pyramid). Indeed, this situation resembles a pattern we observe frequently in the genome as a whole: adjacent domains that are clearly distinct, yet show significant interaction with each other. The fact that the 10A6-7 junction is observed infrequently in polytene band analysis suggests that this interband may be somewhat variable, present on some fraction of chromosomes and absent in others. This is a very interesting possibility that we discuss in the revised manuscript.

Overall, we feel that these issues are consistent with a correspondence between polytene banding patterns and our Hi-C data in this region. A band (10A3) that is virtually invisible in polytene banding analysis is not detected by Hi-C, and a region that is often seen as a single band, and occasionally as separate bands, shows a Hi-C signature that would be expected from a region that existed in both states within a population of nuclei. Meanwhile, all of the interbands that are clearly visible in polytene spreads are clearly visible in Hi-C data.

Finally, the authors should comment on the correspondence between the boundaries and polytene bands. It stands to reason that a method capable of resolving interbands would need to have superb resolution of bands. How well do the boundaries defined in this manuscript fit with previous cytological and HiC reports for major bands? It is an interesting question, not only for its relevance to the question of HiC reproducibility between labs, but because different tissue sources, including salivary glands, tissue culture cells and embryos have been used. Is it possible that different tissues have the same general organization into TADs, but the detailed boundaries of the TADs differ in some cases?

We thank the reviewer for this suggestion, and took a closer look several regions of associated polytene bands and TADs that were previously examined by Eagen et al. In the revised manuscript, we provide a detailed examination of the region of band 22A1-2, which appeared as a single large TAD in Eagen’s data, but which had occasionally been observed to contain an interband. Our data reveal this locus to have a more complex topology, with two large TADs surrounding a middle region containing smaller domains and a fascinating looped region, which together are likely responsible for the observed interband. This analysis is included in Figure 6 in the revised manuscript.

Reviewer #2:The data largely are clear and compelling except for a few issues:1) The ChIP data for insulator binding sites are from 0-12 hour embryos, a much broader developmental window than the two time points examined for the TAD mapping. Thus authors should discuss how this may or may not affect their conclusions. Given that the insulator ChIP data are not from the same restricted developmental time as the TAD maps, it would be useful to discuss how the boundary elements mapped here compare to sites of localization of CHRIZ (Chromator), even though they have been determined in S2 and Kc cells. Zhimulev et al. concluded that CHRIZ binding was the most diagnostic feature of interbands, so this could serve as further confirmation of the coincidence of boundary regions with interbands.

We agree with the reviewer that the differences in the tissues used to generate Hi-C and ChIP data needs to be clarified, and have done so in the revised manuscript. We have also included CHRIZ in our analyses of the molecular features of our TAD boundaries, and found this protein to be a highly diagnostic feature of our boundaries.

2) Please provide color scale bars for the heatmaps of contact probabilities. This will aid readers in comparing Figure 1 and 7. Please provide these also for Figure 3 and the other protein binding figures.

We have included color scale bars in all relevant figures.

3) The looped regions shown in Figure 8 and the corresponding supplemental figures are difficult for readers unfamiliar with Hi-C to visualize. The interaction in Figure 8 is clear, but could the authors provide a clearer graphic to explain the loops in Figure 8? Could they also highlight the diagnostic loop signatures in the supplemental figures?

We have revised Figure 8 and its supplements, and hope that the looping interactions are clearer.

Reviewer #3:The manuscript is interesting but many of the observations, including chromatin 3D organization in Drosophila embryos and association of boundaries with insulator proteins, have been published previously. The following are other more specific concerns:1) Authors should explain how they calculate the resolution of Hi-C data. Because the frequency of interactions decays with distance, authors should keep in mind that resolution will depend on distance. If I understand Table S1 Sample Information in Supplementary file 1 correctly, the authors made 2 libraries from nc12 and 3 libraries from nc14. I used Juicer to process the data for these two libraries and the results are as follows:Experiment description: nc12Sequenced Read Pairs: 93,483,816Normal Paired: 65,205,173 (69.75%)Chimeric Paired: 16,627,312 (17.79%)Chimeric Ambiguous: 6,549,875 (7.01%)Unmapped: 5,101,456 (5.46%)Ligation Motif Present: 45,319,979 (48.48%)Alignable (Normal+Chimeric Paired): 81,832,485 (87.54%)Unique Reads: 26,913,205 (28.79%)PCR Duplicates: 54,884,068 (58.71%)Optical Duplicates: 35,212 (0.04%)Library Complexity Estimate: 28,537,160Intra-fragment Reads: 760,888 (0.81% / 2.83%)Below MAPQ Threshold: 6,649,224 (7.11% / 24.71%)Hi-C Contacts: 19,503,093 (20.86% / 72.47%)Ligation Motif Present: 7,811,696 (8.36% / 29.03%)3' Bias (Long Range): 84% – 16%Pair Type% (L-I-O-R): 25% – 25% – 25% – 25%Inter-chromosomal: 1,545,226 (1.65% / 5.74%)Intra-chromosomal: 17,957,867 (19.21% / 66.73%)Short Range (<20Kb): 8,504,570 (9.10% / 31.60%)Long Range (>20Kb): 9,451,838 (10.11% / 35.12%)Experiment description: nc14Sequenced Read Pairs: 393,928,165Normal Paired: 144,374,310 (36.65%)Chimeric Paired: 24,393,845 (6.19%)Chimeric Ambiguous: 9,682,276 (2.46%)Unmapped: 6,884,964 (1.75%)Ligation Motif Present: 127,838,532 (32.45%)Alignable (Normal+Chimeric Paired): 168,768,155 (42.84%)Unique Reads: 119,180,726 (30.25%)PCR Duplicates: 49,523,419 (12.57%)Optical Duplicates: 64,010 (0.02%)Library Complexity Estimate: 227,824,974Intra-fragment Reads: 1,146,075 (0.29% / 0.96%)Below MAPQ Threshold: 29,415,704 (7.47% / 24.68%)Hi-C Contacts: 88,618,947 (22.50% / 74.36%)Ligation Motif Present: 41,585,911 (10.56% / 34.89%)3' Bias (Long Range): 83% – 17%Pair Type% (L-I-O-R): 25% – 25% – 25% – 25%Inter-chromosomal: 5,442,733 (1.38% / 4.57%)Intra-chromosomal: 83,176,214 (21.11% / 69.79%)Short Range (<20Kb): 49,153,740 (12.48% / 41.24%)Long Range (>20Kb): 34,004,040 (8.63% / 28.53%)This suggests that nc12 has 19 million HiC contacts and nc14 has 88.6 million HiC contacts. It is impossible to have a 500 bp resolution, as the authors claim, with these numbers of reads. The authors state that "We conclude that these Hi-C are of high quality and reproducibility". There are currently two HiC datasets from Drosophila available in Juicebox, and both of them have around 1 billion mapped reads. The HiC libraries made with Drosophila embryos by Hug et al. have around 400 million mapped reads. Therefore, the libraries presented in this study cannot be described as high quality compared to others already available. Furthermore, the nc12 library is of much lower quality than the nc14 library. This is a critical issue when interpreting the results.

As discussed above there is some misunderstanding about our data. Data with our original submission contained > 250,000,000 contacts at nc14 and we have supplemented it bring the total to > 450,000,000. We have removed specific claims about the resolution, since this is a poorly defined concept, and instead let the data speak for itself.

2) Authors should use the dm6 genome instead of dm3.

We agree with the reviewer, and have re-analyzed all sequencing data by mapping to dm6 (and lifting over to dm3 where needed to correspond with published datasets). All relevant genomic coordinates are provided primarily in dm6 (in figures). Because so many existing datasets are in dm3, and because the UCSC browser has substantially richer data for the dm3 genome, we have also provided dm3 coordinates where possible to aid others in using and expanding our work.

3) It is strange that interbands in polytene chromosomes correspond to boundaries. These structures have always been thought to contain active genes. Hug et al. identify very small domain that they also claim are boundaries. In their Hi-C data (see for example Figure 1), once can observed interactions between these small domains that resemble compartmental interactions in mammalian Hi-C data, suggesting that these small domains do not correspond to boundaries, they correspond to active domains. These features are not observed in the Hi-C data presented in this manuscript, which may be due to the fewer reads in the Hi-C data. Authors need to consider this and try to differentiate between effects of transcription and effects of insulator proteins.

The nature of interbands has long been controversial, and it has indeed been proposed that interbands correspond to active genes. A number of pieces of evidence argued against this model, including the identification of gene bodies that reside in polytene bands, dry weight measurements that indicated that interbands represented an insufficient fraction of genomic sequence cover actively transcribed loci, and recently, fine mapping of interbands to smaller genomic regions that largely separate promoters. The Zhimulev 2014 PLoS One paper provides a fairly detailed review of much of this evidence.

We agree with the reviewer that there are additional levels of complexity in Hi-C data. We observe strong compartment effects, as well as hierarchical interactions between domains of various sizes. These are interesting phenomena for future work, and we plan on pursuing many of them. We do not attempt to address all of these potential levels of organization in this manuscript. Rather, we feel that the strong correspondence between the domain boundaries that we have identified and both the molecular features of insulators and locations of mapped polytene interbands make a compelling case that these regions represent a significant contribution to the architecture of fly chromosomes.

4) Authors cannot directly compare features of nc12 and nc14 embryos unless they have the same number of Hi-C contacts in both samples. I think it is critical to obtain more reads for the nc12 libraries.

We agree that these comparisons require deeper complexity nc12 libraries. We included nc12 in the original manuscript because we had the data and thought it would be of interest, but nc12 was not central to any of our conclusions and we have decided not to include it here.